# UNMUTE THE PATCH TOKENS: RETHINKING PROBING IN MULTI-LABEL AUDIO CLASSIFICATION

**Lukas Rauch**[*1] **René Heinrich**[1,2] **Houtan Ghaffari**[3] **Lukas Miklautz**[4] **Ilyass Moummad**[5]
**Bernhard Sick**[1] **Christoph Scholz**[1,2] *[*]lrauch@uni-kassel.de

[1]University of Kassel [2] Fraunhofer IEE [3]Ghent University [4] ML and Systems Biology, MPI of Biochemistry [5]INRIA Montpellier

## ABSTRACT

Although probing frozen models has become a standard evaluation paradigm, self-supervised learning in audio defaults to fine-tuning when pursuing state-of-the-art on AudioSet. A key reason is that global pooling creates an information bottleneck causing linear probes to misrepresent the embedding quality: The `cls`-token discards crucial token information about dispersed, localized events in audio. This weakness is rooted in the mismatch between the pretraining objective (globally) and the downstream task (localized). Across a comprehensive benchmark of 13 datasets and 6 spectrogram-based encoders, we investigate the global pooling bottleneck. We introduce binarized prototypical probes: a lightweight and simple pooling method that learns prototypes to perform class-wise information aggregation. Despite its simplicity, our method notably outperforms linear and attentive probing. Our work establishes probing as a competitive and efficient paradigm for evaluating audio SSL models, challenging the reliance on costly fine-tuning.

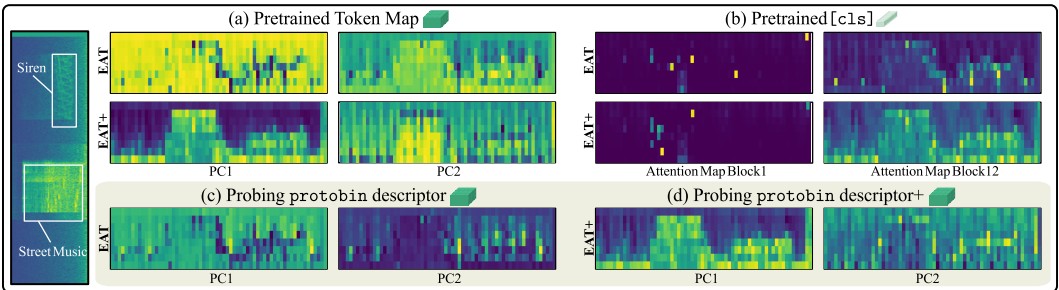

Figure 1: **The pooling bottleneck.** Visualizing embeddings from a purely self-supervised model (EAT) and its supervised[+]-adapted version (EAT[+]) for a spectrogram from `urban`. **(a)** A PCA of the token map shows that EAT embeddings are rich but entangled, a result of the masked prediction objective, while EAT[+] embeddings are localized and aligned with input events. **(b)** The `cls`-token's attention starts similarly for both models, but is diffuse for EAT in later layers, while EAT[+] becomes spatially selective, highlighting its limitation as a probe vector. **(c)** Our `protobin` disentangles these correlated EAT embeddings to recover localized event information. **(d)** For the EAT[+] model, `protobin` further enhances the embeddings, providing a superior representation to the `cls`-token.

## 1 INTRODUCTION

Self-supervised learning (SSL) promises general-purpose embeddings that transfer across downstream tasks (Oquab et al., 2024). A key advantage is their out-of-the-box utility: instead of compute- and label-intensive fine-tuning, one can freeze the pretrained backbone and train only a lightweight probe. As an evaluation paradigm, probing offers a faithful and efficient assessment of pretrained embeddings by minimizing the confounding factors of fine-tuning (Chen et al., 2020; Rauch et al., 2025a). Fine-tuning often yields stronger downstream performance (Park et al., 2023), but can obscure the intrinsic quality of the frozen embeddings (Kumar et al., 2022). While probing is an established evaluation paradigm in computer vision (Oquab et al., 2024; Darcet et al., 2025) and is also utilized in audio SSL (Niizumi et al., 2024; Yadav et al., 2024) on benchmarks such as HEAR (Turian et al., 2022), the pursuit of state-of-the-art (SOTA) performance on AudioSet (Gemmeke et al., 2017) still defaults to resource-intensive fine-tuning (Alex et al., 2025). This discrepancy

motivates our central question: why does this influential benchmark still lack a lightweight probing method that reliably reflects a model's performance as an alternative to fine-tuning?

The performance of a frozen probe depends on the interplay between the *pretraining objective* (i.e., the pretext task) and the *pooling method* (i.e., embedding extraction). Poor probing performance for masked image modeling (MIM)-pretrained models is a direct result of the pooling method, as the `[cls]`-token distributes attention too uniformly instead of focusing on key information (Przewięźlikowski et al., 2025; Alkin et al., 2025). The superior performance of probes that utilize the full token map (Psomas et al., 2025) creates a critical deficit for simpler methods, rendering them unreliable proxies for an encoder's embedding quality. This motivates the need for probes that can efficiently leverage all available information to provide a faithful assessment, avoiding the cost and confounding factors of fine-tuning. Many spectrogram-based audio SSL encoders that report SOTA performance on AudioSet via fine-tuning apply MIM-style objectives, often coupled with student-teacher distillation (Chen et al., 2024; Alex et al., 2025; Ahmed et al., 2024; Chen et al., 2023a; Li et al., 2024). By design, this task induces a bias toward contextualized token-level embeddings, exposing any probe's limitations that collapse the tokens into a simple global summary. While attentive pooling, which learns a token-weighted summary, has emerged as a potential solution in computer vision (Przewięźlikowski et al., 2025), its application to audio remains a research gap, particularly for representing complex polyphonic scenes.

In addition, the *downstream task* plays a role in the performance of probes (Alex et al., 2025). Polyphonic soundscapes are multi-label, with sparse and localized evidence for sound events in the time-frequency domain. Forcing this information into a single descriptor, whether fixed or learnable during probing, creates a single-vector bottleneck: Quieter but important events could be overshadowed by more prevalent sounds, making it difficult for a linear classifier to disentangle the mixed signals (see Figure 1). Therefore, the limited adoption of probing and its failure to approach fine-tuning SOTA performance on AudioSet likely reflects a pooling mismatch, not an absence of usable information. While the pretrained `[cls]`-token struggles to summarize these sparse events and can underperform in audio classification (Alex et al., 2025; Li et al., 2024), fine-tuning implicitly learns a superior, class-conditioned aggregation over the full token map (see Figure 1).

---

**Hypothesis: Pooling Bottleneck**

The limited usage of probing as an evaluation tool for multi-label audio SSL stems from the pooling method. Standard single-vector probes, from the off-the-shelf `[cls]`-token to attentive pooling, underutilize token embeddings. A more valuable and reliable probe requires a shift to per-class, multi-vector aggregation to fully exploit the information in the patch tokens (Figure 2).

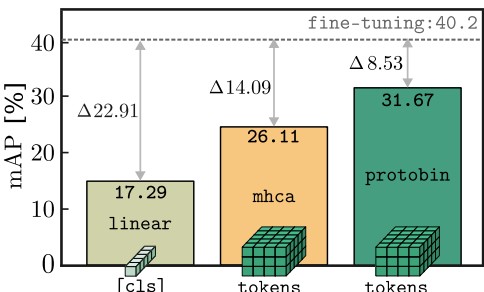

Figure 2: **Probing on `as20k` with EAT.**

---

**Contributions**

1. **Audio probing benchmark.** We conduct an extensive benchmark to systematically investigate the pooling bottleneck in audio SSL. Our analysis establishes a probing hierarchy, demonstrates that the `cls`-token probe and fine-tuning can be unfaithful evaluators of audio SSL models, quantifies the impact of polyphony in probing, and shows that supervised adaption after pre-training alters `cls`-token's quality and model rankings. We empirically show that the bottleneck stems from the pooling method, not the embeddings, challenging the validity of current evaluation practices in achieving SOTA performance on AudioSet.

2. **Elevating probing in audio.** Prototype methods notably outperform other pooling methods, including linear and attentive. This result challenges the reliance on costly fine-tuning and establishes probing as a competitive and efficient paradigm for evaluating audio SSL models.

3. **Binarized prototypical probes.** We introduce an efficient probe that addresses the pooling bottleneck by performing class-wise and multi-vector information aggregation on the tokens. We simplify prior prototypical approaches by decoupling prototypes from class constraints and eliminating an explicit orthogonality loss, while achieving competitive performance.

## 2 PROBING FROZEN EMBEDDINGS IN MULTI-LABEL AUDIO

This section formally introduces the probing task for (multi-label) audio, provides a taxonomy of the pooling methods, and introduces our binarized prototypical probes.

### 2.1 PROBLEM FORMULATION AND NOTATION

We consider a multi-label classification task with a training dataset $\mathcal{D} = \{(x_i, \mathbf{y}_i)\}_{i=1}^{N}$, where each input $x_i$ belongs to a set of spectrograms $\mathcal{X} \subseteq \mathbb{R}^{T \times F}$ with $T$ time frames and $F$ frequency bins. Each corresponding one-hot-encoded target vector $\mathbf{y}_i \in \{0, 1\}^C$ indicates the presence or absence of $C$ possible classes. Multiple classes may simultaneously occur for a single input. Additionally, we assume access to a pretrained embedding encoder $f_\theta$, parameterized by frozen weights $\theta$. This model $f_\theta$ functions as a feature extractor, mapping an input $x_i$ to a token map:

$$\mathbf{z}_i = f_\theta(x_i) \in \mathbb{R}^{D \times S_t \times S_f} \quad (1)$$

where $D$ is the embedding dimension, and $S_t$, $S_f$ index a grid of time and frequency patch tokens. If the backbone exposes a [cls]-token, we denote it by $\mathbf{s}_i^{\text{cls}} \in \mathbb{R}^D$. For instance, a 10-second log-Mel spectrogram with $F=128$ Mel bins (from 16 kHz audio) is patched into non-overlapping $16 \times 16$ time-frequency tokens, yielding $T \approx 1024$ frames and thus $S_t=64$ and $S_f=8$. With an embedding dimension of $D=768$, the resulting token map is $\mathbf{z}_i \in \mathbb{R}^{768 \times 64 \times 8}$. Given the frozen token map $\mathbf{z}_i$, a probe $g_\phi$ consumes a pooled descriptor $\tilde{\mathbf{z}}_i$, determining how information is extracted. The resulting vector is then processed by the probe, typically a linear classifier $g_\phi(\tilde{\mathbf{z}}_i) = \mathbf{W}\tilde{\mathbf{z}}_i + \mathbf{b}$.

### 2.2 A TAXONOMY OF GLOBAL POOLING METHODS

This section provides a brief taxonomy of pooling methods to contextualize our investigation.

**Fixed global pooling (single-vector, non-learnable).** The default approach collapses the token map $\mathbf{z}_i$ from the frozen backbone $f_\theta$ into a single descriptor $\tilde{\mathbf{z}}_i = A(\mathbf{z}_i) \in \mathbb{R}^D$ via a non-learnable aggregator $A : \mathbb{R}^{D \times S_f \times S_t} \to \mathbb{R}^D$, followed by a linear probe. If the model exposes a last-layer [cls]-token $\mathbf{s}_i^{\text{cls}}$, produced via self-attention, one can set $\tilde{\mathbf{z}}_i := \mathbf{s}_i^{\text{cls}}$. While mean pooling all tokens $\tilde{\mathbf{z}}_i$ is a viable alternative, all encoders in our benchmark provide a cls-token, making it our standard for fixed global pooling. A $k$-NN probe is also used in multi-class settings, but vanilla $k$-NN performs single-label majority voting and is ill-suited to multi-label.

**Learnable global pooling (single-vector, learnable).** Instead of a fixed pretext-task descriptor, this pooling family uses a learnable module to aggregate the token map into a single descriptor $\tilde{\mathbf{z}}_i$ while keeping $f_\theta$ frozen. Attentive variants assign data-dependent weights to tokens and form a weighted summary. They typically outperform fixed global pooling for pretrained encoders in computer vision (El-Nouby et al., 2024; Darcet et al., 2025).

### 2.3 LEARNABLE PROTOTYPICAL POOLING: A PER-CLASS POOLING METHOD

As an alternative to single-vector pooling, prototypical probes aggregate evidence per class via multiple learnable exemplars (i.e., prototypes). Inspired by explainability methods (Chen et al., 2019; Donnelly et al., 2022), the idea is to score the frozen token map by its similarity to learnable prototypes in the embedding space, which naturally accommodates dispersed events by allowing different classes to localize information in distinct time-frequency regions (Rauch et al., 2025a).

**Binarized prototypical probes.** We introduce *binarized prototypical probes*, a novel and efficient instance from the prototypical pooling family (Rauch et al., 2025a; Heinrich et al., 2025) that scores token map embeddings by matching them against a small set of prototypes that are binarized on-the-fly. We maintain a set of $C \cdot J$ total learnable, class-agnostic prototypes, with parameters $\tilde{\mathbf{p}}_j \in \mathbb{R}^D$

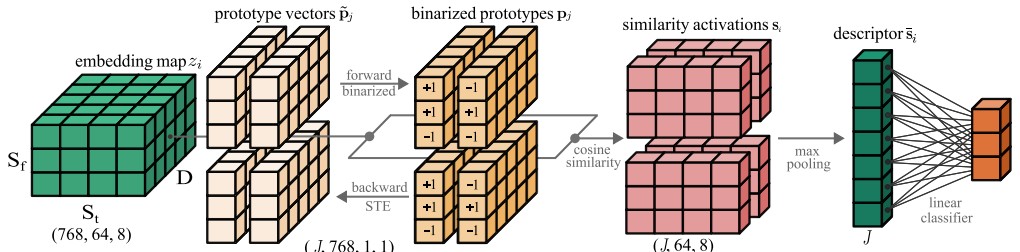

Figure 3: **Binarized prototypical pooling (schematic).** Example shown for a base audio SSL backbone with $D=768$-dim tokens and a $64\times8$ token map. There are $J$ learnable prototypes, which are binarized on-the-fly. Tokens are matched against these prototypes, max pooling aggregates spatial evidence, and a final linear layer maps the resulting prototype scores to class logits.

for each prototype index $j \in \{1, \dots, CJ\}$. At each forward pass, we form the binary prototype

$$\mathbf{p}_j = \text{sign}(\tilde{\mathbf{p}}_j) \in \{-1, +1\}^D. \tag{2}$$

This constraint helps encouraging large angular margins between distinct prototypes. The near-orthogonality is an emergent property, forcing prototypes to the corners of a high-dimensional hypercube, creating a strong structural bias for diversity. The optimization process seeks discriminative features to minimize classification loss and is incentivized to select orthogonal solutions. The non-differentiability of $\text{sign}(\cdot)$ is handled with the straight-through-estimator (STE) (Bengio et al., 2013): during back-propagation, $\frac{\partial \, \text{sign}(x)}{\partial x} \approx 1$, so the forward pass uses hard $\pm 1$ while gradients flow to the real-valued $\tilde{\mathbf{p}}_j$. Given the frozen token map $\mathbf{z}_i = f_\theta(x_i) \in \mathbb{R}^{D \times S_t \times S_f}$, let $\mathbf{z}_i^{t,f} \in \mathbb{R}^D$ denote the token at time-frequency index $(t, f) \in \{1, \dots, S_t\} \times \{1, \dots, S_f\}$. We score each prototype against all tokens using cosine similarity and aggregate evidence via max-pooling:

$$s_j(t, f) := \frac{\mathbf{p}_j^\top \mathbf{z}_i^{t,f}}{\|\mathbf{p}_j\|_2 \, \|\mathbf{z}_i^{t,f}\|_2}, \qquad \bar{s}_j := \max_{t,f} s_j(t, f). \tag{3}$$

Stacking the pooled scores across all $J$ prototypes yields the vector $\bar{\mathbf{s}}_i \in \mathbb{R}^J$. We use this vector as the clip-level descriptor, i.e., set $\tilde{\mathbf{z}}_i := \bar{\mathbf{s}}_i$, and map it to class logits with the linear classifier $g_\phi$.

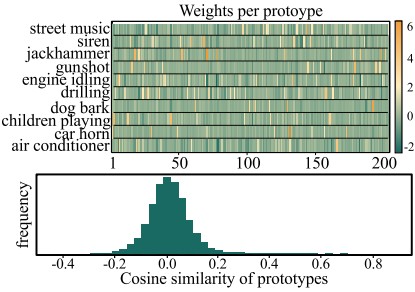

Figure 4: **Weights and similarities example.** Trained `protobin` on `urban`.

**Rationale.** A prototype layer is parameter-efficient, requiring only $J \cdot D$ parameters. The value for $J$ is set by multiplying the number of classes $C$ by a small constant (e.g., 20 (Rauch et al., 2025a)), offering a compact alternative to attentive pooling heads that can require over $2D^2$ parameters (El-Nouby et al., 2024). By binarizing the prototypes to $\mathbf{p}_j \in \{-1, +1\}^D$, our method yields an additional $32\times$ memory reduction relative to 32-bit floats, making it ideal for memory-constrained and on-device applications (e.g., bioacoustics). Cosine matching inherently keeps scores scale- and dimension-invariant across backbones. The near-orthogonality observed between prototypes (cf. Figure 4) is not enforced by an explicit mechanism but is an emergent property arising from the method. Through the binarization, we also constrain them to the corners of a $D$-dimensional hypercube, creating a structural bias for diversity. During training, the optimization process seeks a set of maximally discriminative prototypes to effectively classify different audio events. The optimization process, seeking to minimize classification loss, is incentivized to select distinct, non-redundant prototypes. In the high-dimensional embedding space, this is most effectively achieved when prototypes are nearly orthogonal. Therefore, we simplify the training objective by eliminating the need for an explicit orthogonality loss term used in prior work to enforce diversity (Rauch et al., 2025a; Heinrich et al., 2025). Finally, unlike prior work, we make the prototypes class-agnostic, allowing the prototypes to better collaborate in disentangling task information. The link between these diverse prototypes and their specific class contributions is learned entirely by the linear classifier. This layer learns to map the similarity scores from the $J$ prototypes to the $C$ class logits, effectively assigning semantic meaning to each prototype based on its utility for the classification task.

## 3 RELATED WORK

**SSL paradigms in audio.** In vision, two families dominate modern SSL: student-teacher clustering/distillation (Caron et al., 2021; 2020) and MIM (He et al., 2022; Darcet et al., 2025). Hybrids combining global invariance with masking are considered the current best-performing models (Oquab et al., 2024; Assran et al., 2023). Spectrogram-based audio SSL largely adapts these paradigms: vision transformer (ViT) backbones trained via masked-spectrogram prediction or student–teacher paradigms with audio-specific augmentations (see Table 1). Audio-MAE (Huang et al., 2022) and Dasheng (Dinkel et al., 2024) are generative masked reconstruction models (He et al., 2022). BEATs (Chen et al., 2023a) follows BEiT-style masked token prediction with discrete acoustic tokenizers (Bao et al., 2022). ASiT (Ahmed et al., 2024), EAT (Chen et al., 2024), and SSLAM (Alex et al., 2025) use momentum-teacher distillation with masked/local–global or utterance–frame objectives (Caron et al., 2021; Baevski et al., 2022). Except for Dasheng (which uses additional datasets), these models pretrain on AudioSet's `as2m` (Gemmeke et al., 2017), establishing an influential line of work where SOTA is measured mostly by fine-tuning performance.

Table 1: **Spectrogram-based backbones used in our work.** Mask: input masking during pretraining. EMA: student–teacher with EMA teacher. Global [`cls`]: explicit global/token objective during pretraining. Supervised[+] have an additional fine-tuned checkpoint on `as2m` available.

| Model | Year | Paradigm | Supervised[+] | Mask | EMA | Global [`cls`] | Pretrain data |
|-------|------|----------|:---:|:---:|:---:|:---:|:---:|
| A-MAE | 2022 | Masked spec reconstruction | ✗ | ✓ | ✗ | ✗ | as2m |
| BEATs | 2022 | Masked token prediction | ✓ | ✓ | ✗ | ✗ | as2m |
| ASiT | 2024 | Masked spec reconstruction + latent distillation | ✗ | ✓ | ✓ | ✓ | as2m |
| EAT | 2024 | Masked latent distillation | ✓ | ✓ | ✓ | ✓ | as2m |
| Dasheng | 2024 | Masked spec reconstruction | ✗ | ✓ | ✗ | ✗ | as2m[*] |
| SSLAM | 2025 | Masked latent distillation + mixtures | ✓ | ✓ | ✓ | ✓ | as2m |

**Evaluation in audio SSL.** Simple linear probes are widely used in computer vision (Oquab et al., 2024) and utilized by numerous audio SSL works (Niizumi et al., 2022; 2024; 2025; Yadav et al., 2024; Li et al., 2024; Pepino et al., 2025) on benchmarks such as HEAR (Turian et al., 2022) with a simple probing toolkit. However, these evaluations in audio SSL have largely treated probing as a fixed protocol. With the notable exception of a token reshaping approach from Niizumi et al. (2022), the impact of the pooling method and the underlying performance bottleneck it creates, has remained largely unexplored. When pursuing SOTA performance on AudioSet, audio SSL still defaults to fine-tuning (Huang et al., 2022; Chen et al., 2023a; Ahmed et al., 2024; Chen et al., 2024; Alex et al., 2025). We attribute this reliance on fine-tuning, further justified by the sentiment that linear probes cannot fully reflect embedding quality (Li et al., 2024), to a pretext-pooling mismatch: pretraining learns token-level information, yet standard probes compress the tokens into a single global vector, discarding per-source cues critical for polyphony and localized events. A-MAE yields weak linear probe utility in bioacoustics (Rauch et al., 2025b), consistent with findings that generative objectives disperse information across tokens (Park et al., 2023; Alkin et al., 2025). This limitation becomes evident in the line of work pursuing SOTA on AudioSet: Although masked-distillation models (BEATs, EAT, ASiT) are designed to produce stronger global representation in the [`cls`]-token, their performance with frozen-backbone probing is rarely reported in related work. SSLAM includes linear probing results on selected datasets, yet cross-backbone comparability is limited (Alex et al., 2025). Dasheng reports frozen MLP and $k$-NN results on multi-class tasks on HEAR but does not address multi-label settings. This gap motivates a systematic study of probing methods for frozen audio embeddings.

**Advanced probe architectures.** Replacing fixed global pooling with learned pooling over token maps during probing improves alignment with MIM (Darcet et al., 2025). Attentive pooling consistently outperforms fixed global pooling (Psomas et al., 2025; El-Nouby et al., 2024; Darcet et al., 2025). Complementary analyses show that [`cls`] attention of backbones tends to be diffuse under MIM, weakening it as a global descriptor (Przewięźlikowski et al., 2025). Some works in audio have explored structured, non-attentive pooling. For instance, Niizumi et al. (2022) utilize the token map by concatenating frequency features at each time step before temporal pooling (`linpre`). Attentive pooling methods compute token weights and values differently, ranging from single-query multiple-instance learning (`abmilp`) (Ilse et al., 2018) and multi-head cross-attention (`mhca`) (El-Nouby et al., 2024; Chen et al., 2023b; Bardes et al., 2024) to data-dependent single-head (`simpool`) (Psomas

et al., 2023) and efficient multi-query (ep) approaches (Psomas et al., 2025). Other work in audio explores learnable per-class prototypes for probing (Rauch et al., 2025a), matched to multi-label audio where classes localize in distinct time-frequency regions. Real-valued prototype probes show promising results in bioacoustics (Rauch et al., 2025a). Token-aware attention and prototype designs better align with MIM embeddings and polyphonic labels than single-vector summaries, yet evaluations in audio SSL remain sparse. This further motivates our comprehensive analysis.

**Positioning of this work.** Prototype layers originated in vision for interpretability (Chen et al., 2019; Donnelly et al., 2022) and were adapted to bioacoustics (Heinrich et al., 2025). Closest to our setting, Rauch et al. (2025a) apply prototypical probing over spectrogram tokens for a domain-specific MAE in bird sound classification. We extend this line of research with a binarized STE variant that constrains prototypes to the hypercube, yielding strong compression and margin-like regularization. Additionally, our work introduces two key architectural simplifications. First, we decouple prototypes from classes, allowing class-agnostic features to emerge automatically via the final linear layer. Second, we find that the supervised learning signal is sufficient for prototype diversity in this context, eliminating the need for an explicit orthogonality loss term (Rauch et al., 2025a). Our variant of these simplified prototypes remains highly competitive while offering a 32x memory reduction, consistent with successes of discrete parameterizations (Courbariaux et al., 2015; Hubara et al., 2016). Beyond these method-level contributions, our work establishes prototypical probing as a general evaluation paradigm for the audio SSL field and delivers the first extensive probing benchmark. This study adapts recent attentive methods from vision to serve as strong baselines and ultimately reveals a clear hierarchy. While learned pooling is broadly advantageous, mirroring trends in vision (Psomas et al., 2025; Darcet et al., 2025), prototypical methods consistently set the SOTA results for probing in audio SSL, providing a competitive alternative to fine-tuning.

## 4 EXPERIMENTAL STUDY: A BENCHMARK ON PROBING IN AUDIO

This section first outlines our experimental setup, including backbones, pooling methods, datasets and the evaluation protocol of the benchmark. It is followed by our main results organized as focused questions with short rationales.

### 4.1 EXPERIMENTAL SETUP AND EVALUATION PROTOCOL

**Backbones.** We evaluate six state-of-the-art frozen spectrogram-based SSL encoders $f_\theta$, summarized in Table 1. To ensure a fair comparison, we only use the ViT-base checkpoints with an embedding dimension $D$ of 768 and circa 86M parameters since this is the only configuration offered across all models. We also include supervised⁺-checkpoints that were fine-tuned on as2m in addition to pretraining. Such variants exist for EAT, BEATs, and SSLAM. Reporting results for the purely self-supervised and the supervised⁺ versions allows us to quantify how supervised adaptation to the AudioSet label space affects the quality of frozen embeddings (see Figure 1).

**Datasets.** We organize the benchmark into three topical groups. The primary group, general multi-label audio, contains the smaller, balanced AudioSet subset as20k (Gemmeke et al., 2017) and fsd50k (Fonseca et al., 2022), a curated dataset aligned with the AudioSet ontology. Following Alex et al. (2025), we also include the polyphonic datasets desed (domestic sound events with 10 labels) (Johnson et al., 2021), spass (urban soundscapes with 28 labels (Viveros-Muñoz et al., 2023)), and urban (urban soundscapes with 10 labels (Salamon et al., 2017)). The second group focuses on fine-grained multi-label bioacoustics, for which we use seven subsets from the birdset benchmark (Rauch et al., 2025b). These tasks test the models' generalization under a domain shift and a data-efficient, 64-shot few-shot learning protocol. The third group provides multi-class datasets using the esc50 and sc-2 datasets. These single-label tasks serve as a control condition to isolate the impact of polyphony and determine whether the pooling bottleneck is unique to the multi-label audio setting. Appendix D.1 provides a detailed description of each dataset.

**Pooling methods.** We compare eleven pooling methods (cf. Section 3) that operate on frozen encoders. Each technique produces a descriptor $\tilde{z}$ that is passed through a linear classification layer. Linear and mlp consume only the fixed global [cls]-token as a compact summary of the input. Linearc, conv use the token map without attention. The former concatenates all tokens to form the descriptor. The latter applies a lightweight convolution for local aggregation. Linpre also utilizes the token map by concatenating frequency features at each time step before temporal pooling. Atten-

tive pooling of the token map include `abmilp`, `simpool`, `ep`, and `mhca` (see Section 3). Prototypical pooling for class-conditioned descriptors includes the class-dependent `proto` (Rauch et al., 2025a) and our class-agnostic `protobin` (see Section 2.3). Refer to Appendix D.3 for a detailed overview.

**Caching and probing.** For each input $x_i$ in a dataset, we run an augmentation-free forward pass through the encoder $f_\theta$ and cache embeddings from the final hidden block. It contains the full token map $\mathbf{z_i} \in \mathbb{R}^{D \times S_f \times S_t}$ and the fixed global descriptor $\tilde{\mathbf{z}}_\mathbf{i} \in \mathbb{R}^D$ given by the last-layer's [cls]-token $\mathbf{s}_i^{\texttt{cls}}$. This produces a static on-disk embedding store per backbone that we use as input to all probes. Caching avoids repeated model inference and isolates embedding quality at the cost of a less diverse training distribution with on-the-fly data augmentations. We accept this trade-off to preserve computational efficiency as one of the central advantages of probing.

**Training setup.** All probes are trained for 30 epochs with AdamW, a cosine-annealed learning rate scheduler (Loshchilov & Hutter, 2017), a batch size of 128, and the asymmetric multi-label loss (Ridnik et al., 2021). This setup ensured convergence in preliminary studies across probing methods. We apply the default settings to all pooling methods. For prototypical pooling methods, the prototype learning rate equals the global learning rate, and the number of prototypes $J$ is fixed at 20 per class across datasets (10 for `as20k`), following Rauch et al. (2025b). While this fixed value ensures a fair comparison across pooling methods without confounding factors, we provide a sensitivity analysis in Appendix B which confirms that $J$=20 prototypes is a robust choice for our benchmark. For future work, we hypothesize that $J$ could be tuned for specific applications based on factors such as the intra-class diversity and the degree of polyphony in a given dataset.

**Hyperparameter selection.** To keep comparisons fair, we optimize only two scalars: learning rate and weight decay. For each dataset, we select hyperparameters on a validation split and report final results on the held-out test split. If a validation split is unavailable, we reserve 20% of the training set. For datasets with $F$-fold cross-validation, we designate one fold a priori for hyperparameter search. We use a two-stage procedure per {backbone, dataset, probe}-combination. First, we run 50 trials with a fixed seed for comparability, using Sobol (Sobol, 1998) exploration for the first 25 trials and TPE for the remainder (Bergstra et al., 2011), under a successive-halving schedule. All other training details are held constant across probes and backbones. Second, we take the top-$k$ configurations and re-evaluate each with five random seeds to estimate the mean and standard deviation of the validation set's mean average precision (mAP). We then choose the configuration with the highest mean mAP, retrain it with this setting, and evaluate on the test set. Appendix D.4 provides more details on hyperparameters.

## 4.2 EXPERIMENTAL RESULTS

To investigate the pooling bottleneck hypothesis, we conduct an extensive benchmark with focused questions. Our primary analysis evaluates all ten pooling methods across five general multi-label datasets using six encoders and their three supervised⁺-adapted versions. For more targeted analyses, we use a representative subset of the most informative probes to test our hypothesis on seven fine-grained, few-shot bioacoustic datasets and two multi-class control tasks. Throughout our experiments, we report mean average precision (mAP) for multi-label tasks and accuracy for multi-class tasks. The complete results, which form the basis for all visualizations, are detailed in Appendix A.

**(Q$_1$) Pooling hierarchy:** *Is there a best-performing pooling method?*

**Rationale:** A clear hierarchy with prototypes outperforming single-vector methods would support our hypothesis that probing benefits from multi-vector aggregation.

**(Q$_1$) Takeaway.** Across backbones and datasets, the pairwise win matrices in Figure 5, supported by the absolute results in Table 2, show a stable and strong hierarchy. Our `protobin` wins most often, with an average improvement of +14.41 $\%_p$ mAP over `linear` on general audio and +12.16 $\%_p$ mAP on few-shot bioacoustics. Attentive pooling, with `mhca` as the best-performing attentive method, and the simple, reshaping-based `linpre` improve over fixed global descriptors but still lag behind prototypes (-4.59 $\%_p$ mAP) on general audio despite its complexity. Simple baselines are at the bottom, including [cls]-token probes and naive token concatenation (`linearc`). This clear ordering provides strong support for our pooling bottleneck hypothesis: global single-vector probes severely underutilize the rich information in the token map, and a reliable evaluation of current MIM-based audio SSL models benefits from a shift to per-class, multi-vector aggregation. Finally, our results reveal a trade-off between the float-based, class-dependent `proto` and our class-agnostic `protobin`. The full precision and class dependency of

`proto` appears advantageous in specific cases where capturing fine-grained details is critical (e.g. on polyphonic `urban` or with the ASiT backbone). `Protobin`'s simplification makes it a more robust choice for general-purpose evaluation. To help disentangle these architectural factors from the effect of binarization, we provide an ablation with a float-based, class-agnostic variant in Appendix B.

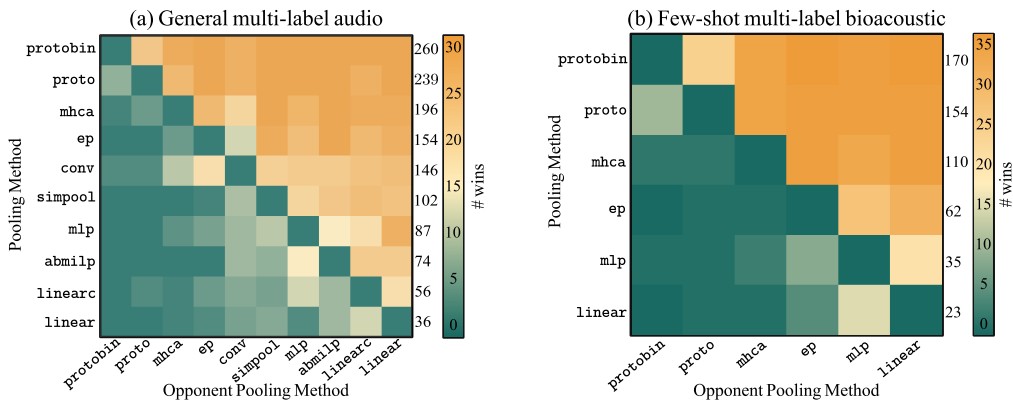

Figure 5: **Pairwise win matrices for pooling methods.** Each cell shows the number of configurations where a method outperforms another (ties omitted, one sd above opponent), aggregated over all datasets and base (non-supervised⁺) backbones. Extracted from Table 2 and Table 5 (Appendix D.2).

Table 2: **Probing benchmark results in general audio.** All results are the mean with std reported in mAP, averaged over five seeds. Best and second best probe per (dataset, backbone) are highlighted.

| Input | | [cls] Baseline | | Token Map | | | Token Map (Att.) | | | | Token Map (Prot) | |
|---|---|---|---|---|---|---|---|---|---|---|---|---|
| | Backbone | linear | mlp | linearc | conv | linpre | mhca | ep | simpool | abmilp | proto | protobin |
| as20k | A-MAE | 8.36±0.0 | 8.77±0.3 | 9.66±0.2 | 11.87±1.1 | 16.49±0.1 | 17.09±0.2 | 17.03±0.1 | 14.69±0.0 | 14.24±0.9 | 21.61±0.3 | **22.32**±0.1 |
| | ASiT | 18.35±0.0 | 19.16±0.1 | 13.36±0.1 | 13.80±0.2 | 18.53±0.0 | 18.72±0.2 | 18.95±0.1 | 18.04±0.0 | 16.10±0.5 | 21.89±0.1 | 20.96±0.0 |
| | Dasheng | 20.98±0.1 | 21.09±0.1 | 18.23±0.1 | 18.57±1.1 | 23.56±0.0 | 27.49±0.1 | 26.53±0.1 | 20.89±0.0 | 22.96±1.9 | 27.59±0.1 | 29.94±0.2 |
| | BEATs | 24.71±0.0 | 26.29±0.1 | 15.70±0.0 | 12.80±1.1 | 18.59±0.0 | 21.86±0.2 | 20.81±0.4 | 14.99±0.1 | 12.52±1.9 | 30.54±0.1 | 31.54±0.1 |
| | EAT | 17.29±0.0 | 20.59±0.2 | 21.94±0.0 | 19.50±0.3 | 26.49±0.0 | 26.11±0.2 | 26.83±0.0 | 25.15±0.0 | 19.91±3.4 | 31.06±0.0 | 31.67±0.1 |
| | SSLAM | 17.04±0.0 | 19.99±0.1 | 20.51±0.1 | 17.45±0.5 | 24.81±0.0 | 24.45±0.2 | 25.49±0.0 | 22.59±0.1 | 18.91±4.4 | 30.84±0.0 | 30.94±0.1 |
| fsd50k | A-MAE | 19.71±0.0 | 21.34±0.4 | 25.17±0.7 | 40.59±0.8 | 36.08±0.1 | 45.17±0.5 | 43.23±0.1 | 34.89±0.1 | 32.73±4.3 | 49.65±0.2 | 49.69±0.4 |
| | ASiT | 39.57±0.1 | 41.89±0.3 | 9.87±0.5 | 38.23±0.8 | 39.57±0.1 | 42.28±0.3 | 41.76±0.1 | 37.78±0.1 | 39.59±3.5 | 48.25±0.1 | 46.70±0.2 |
| | Dasheng | 38.08±0.2 | 39.56±0.2 | 37.74±0.5 | 48.88±0.8 | 45.11±0.1 | 52.95±0.2 | 52.44±0.0 | 43.94±0.0 | 43.79±3.5 | 55.23±0.1 | 57.31±0.0 |
| | BEATs | 46.89±0.0 | 49.58±0.3 | 36.35±0.1 | 37.19±1.6 | 39.93±0.0 | 48.51±0.3 | 46.16±0.1 | 40.20±0.0 | 40.32±3.2 | 57.17±0.1 | 58.27±0.2 |
| | EAT | 36.39±0.0 | 44.82±0.1 | 38.36±0.3 | 46.64±0.5 | 48.21±0.1 | 51.06±0.3 | 51.29±0.1 | 49.38±0.1 | 45.93±4.4 | 56.07±0.1 | 55.64±0.4 |
| | SSLAM | 36.06±0.0 | 44.26±0.2 | 37.21±0.4 | 43.50±1.4 | 46.11±0.0 | 51.48±0.5 | 50.83±0.1 | 49.86±0.2 | 46.38±2.4 | 56.93±0.1 | 56.99±0.1 |
| desed | A-MAE | 57.46±0.0 | 60.52±0.1 | 60.88±0.1 | 84.10±0.3 | 71.28±4.2 | 83.57±0.2 | 80.13±0.1 | 72.05±0.0 | 76.69±0.3 | 84.11±0.1 | 85.57±0.1 |
| | ASiT | 72.92±0.0 | 74.19±0.2 | 57.49±0.1 | 81.59±0.2 | 74.91±0.1 | 79.50±0.4 | 76.66±0.0 | 73.57±0.0 | 76.58±0.5 | 82.08±0.2 | 81.74±0.2 |
| | Dasheng | 68.39±0.0 | 68.76±0.1 | 72.48±0.0 | 85.32±1.0 | 74.49±0.6 | 84.53±0.1 | 82.74±0.0 | 75.40±0.0 | 76.48±4.5 | 85.90±0.1 | 86.14±0.3 |
| | BEATs | 77.56±0.0 | 80.56±0.2 | 72.23±0.0 | 86.83±0.6 | 76.97±0.0 | 86.91±0.0 | 81.88±0.0 | 81.08±0.1 | 81.77±1.0 | 89.04±0.1 | 89.22±0.6 |
| | EAT | 76.15±0.0 | 80.92±0.0 | 77.90±0.1 | 86.68±0.3 | 81.00±0.0 | 86.06±0.2 | 84.13±0.1 | 83.43±0.0 | 78.80±5.6 | 88.70±0.1 | 88.82±0.1 |
| | SSLAM | 72.49±0.0 | 77.96±0.1 | 76.82±0.2 | 85.55±0.3 | 80.31±0.0 | 85.44±0.1 | 83.77±0.0 | 83.59±0.0 | 81.69±0.7 | 87.69±0.2 | 88.33±0.3 |
| spass | A-MAE | 58.94±0.0 | 60.56±0.1 | 69.01±0.7 | 80.04±0.8 | 77.08±0.2 | 79.24±0.1 | 71.01±0.4 | 69.84±0.0 | 68.75±0.2 | 78.92±0.2 | 79.95±0.6 |
| | ASiT | 68.80±0.0 | 70.27±0.2 | 46.44±4.5 | 73.26±1.1 | 73.88±0.0 | 75.76±0.5 | 69.44±0.0 | 69.04±0.0 | 68.36±0.6 | 73.66±0.1 | 74.69±0.2 |
| | Dasheng | 66.89±0.0 | 64.07±0.2 | 76.76±0.5 | 75.05±0.7 | 72.07±0.0 | 80.71±0.3 | 73.62±0.0 | 74.16±0.0 | 72.02±0.0 | 76.64±0.2 | 80.93±0.5 |
| | BEATs | 74.22±0.0 | 75.97±0.1 | 79.91±0.5 | 84.81±1.5 | 82.12±0.0 | 83.98±0.2 | 76.61±0.1 | 75.58±0.0 | 69.38±0.3 | 87.76±0.2 | 85.77±0.4 |
| | EAT | 65.96±0.0 | 71.55±0.2 | 84.49±0.0 | 79.15±0.6 | 81.83±0.0 | 83.95±0.3 | 77.35±0.0 | 76.55±0.0 | 64.44±0.0 | 83.09±0.8 | 85.64±0.3 |
| | SSLAM | 68.28±0.0 | 73.05±0.0 | 83.06±0.3 | 79.43±1.8 | 80.74±0.2 | 83.45±0.3 | 76.58±0.0 | 76.09±0.0 | 72.42±1.4 | 85.90±0.4 | 86.01±0.1 |
| urban | A-MAE | 58.72±0.0 | 58.97±0.2 | 40.53±1.2 | 85.28±0.2 | 79.01±0.1 | 82.49±0.2 | 79.83±0.2 | 76.21±0.1 | 73.07±2.5 | 83.63±0.2 | 85.17±0.3 |
| | ASiT | 77.53±0.0 | 77.55±0.2 | 44.53±3.9 | 82.12±0.5 | 79.32±0.0 | 79.93±0.3 | 78.48±0.0 | 77.25±0.1 | 76.76±1.6 | 82.35±0.2 | 82.28±0.2 |
| | Dasheng | 69.61±0.1 | 69.07±0.2 | 75.80±0.1 | 85.76±0.6 | 77.30±0.0 | 84.59±0.2 | 82.31±0.1 | 79.04±0.1 | 77.28±0.8 | 85.97±0.3 | 86.55±0.1 |
| | BEATs | 82.54±0.1 | 83.76±0.0 | 75.90±0.1 | 85.57±0.5 | 81.61±0.0 | 86.23±0.2 | 84.31±0.1 | 82.74±0.0 | 77.89±1.1 | 89.04±0.1 | 88.74±0.2 |
| | EAT | 77.76±0.0 | 81.58±0.1 | 78.45±0.1 | 86.35±1.1 | 84.04±0.0 | 86.43±0.0 | 85.40±0.0 | 83.58±0.1 | 79.93±2.0 | 89.11±0.1 | 89.24±0.2 |
| | SSLAM | 75.86±0.0 | 80.64±0.1 | 77.97±0.1 | 86.23±1.5 | 79.01±0.1 | 86.45±0.3 | 84.87±0.0 | 83.21±0.0 | 80.12±1.6 | 88.82±0.2 | 89.05±0.4 |

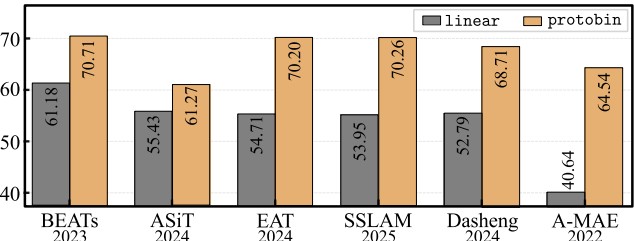

Figure 6: **Backbone averages.** Mean performance across general audio datasets for `linear` and `protobin`. Publication years highlight how probing re-ranks models

**(Q₂) [cls]-token quality.** *Is the `linear` probe a faithful evaluator?*

**Rationale.** We test if the off-the-shelf `linear` probe is a reliable and faithful proxy for embedding quality in audio SSL. A flawed proxy both underestimates the absolute potential of the embeddings and distorts the relative ranking of different backbones.

**(Q₂) Takeaway.** Probing the `[cls]`-token with `linear` is not just a performance bottleneck, it is also an unreliable proxy for pretrained embedding quality in audio SSL. First, Figure 6 shows that the backbone ranking under `linear` is completely reshuffled when using `protobin`. For instance, the backbone ranking is completely inverted: ASiT (from 2024), which appears to be the second-strongest model under `linear`, drops to last place when evaluated with `protobin`. Conversely, the supposedly mediocre SSLAM (current fine-tuning SOTA from 2025), a mid-tier performer with `linear`, is revealed to be a top-tier model, jumping to second place. This demonstrates that the `[cls]`-token is a poor indicator of the model's true token-level embedding quality. Figure 7 confirms this is a systemic issue: `linear/mlp` act as a performance ceiling, and the gains unlocked by token-aware pooling methods vary by backbone. Second, the `[cls]`-token underestimates the true potential of the embeddings. On `as20k`, `protobin` closes 63% of the performance gap to fine-tuning (see Figure 2), demonstrating how much information standard probes discard. This trend holds across all encoders (Table 3), establishing that better pooling provides a more faithful measure of embeddings.

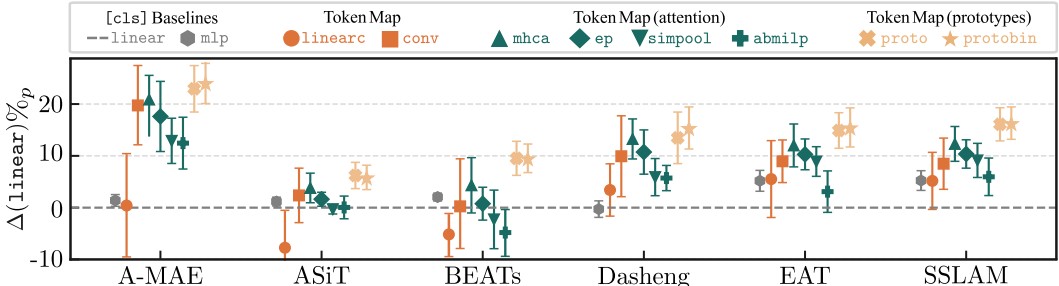

Figure 7: **Performance differences of probes across backbones.** For each backbone, the plot displays the mean and standard deviation of each pooling method as absolute percentage points [%ₚ] compared to the baseline performance of `linear`. All results are extracted from Table 2.

**(Q₃) Multi- vs. single-label.** *Is the pooling bottleneck specific to multi-label?*

**Rationale.** If fixed global pooling degrades from single- to multi-label while token-aware methods remain stable, it would implicate a polyphony-induced bottleneck.

**(Q₃) Takeaway.** On single-label control tasks (`sc-2`, `esc50`), a substantial performance gap persists between the `[cls]`-token probe and token-aware methods, indicating the bottleneck is a general issue of the encoders (Table 3). However, the `[cls]`-probe's performance degrades more sharply than other methods when moving to the multi-label `as20k`. In this single-label setting, `mhca` is often competitive with, or even superior to, our `protobin` probe. This suggests that a well-learned single-vector descriptor can be as effective as our multi-vector approach for single-source audio. This dynamic changes in the presence of multiple sound sources, confirming our core hypothesis. The constant superiority of `protobin` on the multi-label `as20k` task highlights the fundamental limitation of single-vector methods in polyphonic scenes. Methods like `mhca` must compress localized evidence for multiple distinct events into a single vector. In contrast, our multi-vector prototypical approach can activate different specialized prototypes for different sound events within the same audio clip. The discriminative nature of the prototypes is particularly effective at disentangling these overlapping audio events.

Table 3: **Multi- vs. single-label pooling and fine-tuning.** Accuracy on `sc-2` and `esc50` (single-label) and mAP on `as20k` (multi-label). `FT` denotes the reported fine-tuning performance in the respective backbone paper, **bold** marks the best probe per backbone and dataset.

| Backbone | sc-2 (single-label) | | | | esc50 (single-label) | | | | as20k (multi-label) | | | |
|---|---|---|---|---|---|---|---|---|---|---|---|---|
| | linear | mhca | protobin | FT | linear | mhca | protobin | FT | linear | mhca | protobin | FT |
| A–MAE | 12.4 | **84.9** | 79.5 | 98.3 | 22.1 | **86.3** | 83.7 | 94.1 | 8.4 | 17.1 | **22.3** | 37.1 |
| ASiT | 62.2 | 86.3 | **89.5** | 98.9 | 76.1 | 78.3 | **80.3** | 95.3 | 18.4 | 18.7 | **21.0** | 38.6 |
| BEATs | 87.0 | 95.0 | **96.5** | 98.3 | 78.9 | 83.2 | **84.1** | 95.6 | 24.7 | 21.9 | **31.5** | 38.9 |
| EAT | 69.1 | **93.2** | 90.4 | 98.3 | 75.3 | **89.8** | 86.8 | 95.9 | 17.3 | 26.1 | **31.7** | 40.2 |
| SSLAM | 64.8 | **93.8** | 91.9 | 98.1 | 74.2 | **89.0** | 84.7 | 96.2 | 17.0 | 24.4 | **30.9** | 40.9 |

**(Q₄) Takeaway.** On in-domain, general audio tasks (Figure 8a), the probe rankings change notably. The `[cls]`-token-based methods see the largest gains, with `linear` jumping from rank #10 to #6 and `mlp` from #7 to #3. This confirms that supervised fine-tuning injects class-specific information into the global token. Meanwhile, attentive pooling methods are stable, and the prototypical methods retain their top-ranked positions. In contrast, on out-of-domain bioacoustics tasks (Figure 8b), the complete hierarchy remains stable. Despite a minor performance uplift across the board (see

> **(Q₄) Supervised⁺ weights.** *Does extra fine-tuning enrich the token map?*
>
> **Rationale.** Supervised⁺ adaptation injects class information into the `[cls]`-token. This lets us test if the model has learned richer token-level information or just a stronger global descriptor. A localized improvement, where gains are specific to `[cls]` probes and in-domain data, would suggest the latter.

Appendix D.2), the overall ranking is preserved: `linear` remains at the bottom while `protobin` stays at the top. This divergence demonstrates that supervised⁺ primarily strengthens the single-vector `[cls]` descriptor for in-domain tasks but fails to add transferable, token-level information for out-of-domain tasks. The consistent superiority of prototypical methods in both settings further highlights the robustness of per-class, multi-vector aggregation.

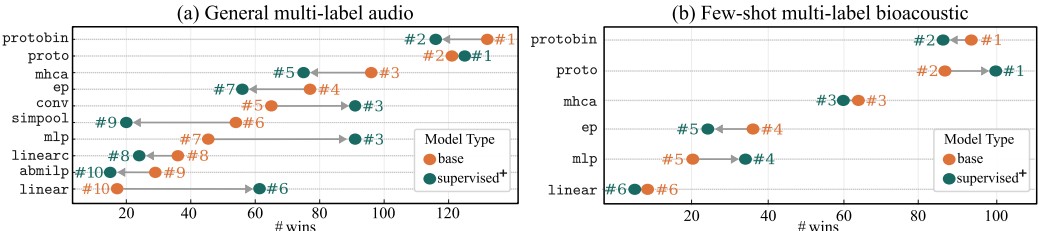

Figure 8: **Pairwise win ranking changes from base to supervised⁺ models.** We display the number of pairwise wins averaged over the backbones with fine-tuned variants (BEATs, EAT, SSLAM) and datasets for each pooling method. Extracted from Table 2 and Table 5 (Appendix D.2).

> **Pooling bottleneck.** Our findings confirm our hypothesis and its implications for probing as a reliable evaluation tool. The `[cls]`-token is a performance bottleneck, underutilizing the token map and leading to an unreliable evaluation (**Q₂**). While attentive pooling offers improvements, our results show multi-vector, per-class aggregation is a more robust strategy, particularly in polyphonic scenes where single-vector methods are limiting (**Q₁, Q₃**). This conclusion holds even when the `[cls]`-token is enhanced by supervised⁺ (**Q₄**). Thus, the primary obstacle to using probing as an evaluation tool is not the quality of the embeddings, but the limitation of the pooling method.

## 5 CONCLUSION AND FUTURE WORK

**Conclusion.** We demonstrated that the underperformance of probing in (multi-label) audio stems not from the frozen embeddings themselves, but from an information bottleneck in pooling methods. Single-vector representations, whether from a fixed `[cls]`-token or learned via attention, are ill-suited for polyphonic audio, as they compress sparse, localized events into a single descriptor. Addressing this, we introduced binarized prototypical probes, a lightweight method that performs per-class aggregation directly on the token map. Our comprehensive benchmark shows this approach consistently outperforms single-vector probes and notably narrows the gap to fine-tuning. By enabling class-conditional vectors with a minimal memory footprint, this work establishes prototypical probing as a viable, efficient, and faithful evaluation paradigm for audio SSL. This challenges the default reliance on costly and confounding fine-tuning when pursuing SOTA on AudioSet.

**Future Work.** A next step is to move beyond the final encoder layer and explore multi-layer feature aggregation, which could unlock even richer embeddings. Furthermore, our token-aware probing framework could be extended from clip-level classification to more granular tasks such as event detection and localization, where the benefits of multi-vector aggregation may be even stronger. While our study focused on audio, the insights into pooling bottlenecks likely apply to other domains as well. Future work could also explore integrating on-the-fly data augmentations with a frozen backbone to push the performance ceiling of the probing paradigm even higher.

## Author Contributions

**Lukas Rauch:** Conceptualization (Lead), Methodology (Lead), Software (Lead), Investigation (Lead), Writing – Original Draft (Lead), Validation (Lead), Visualization (Lead).

**Rene Heinrich:** Conceptualization (Supporting), Validation (Support), Writing – Review & Editing.

**Houtan Ghaffari:** Validation (Support), Visualization (Support), Writing – Review & Editing.

**Lukas Miklautz:** Conceptualization (Supporting), Writing – Review & Editing.

**Ilyass Moummad:** Conceptualization (Supporting), Writing – Review & Editing.

**Bernhard Sick:** Funding Acquisition, Resources.

**Christoph Scholz:** Conceptualization (Supporting), Funding Acquisition, Writing – Review & Editing.

## Ethics Statement

Our research is conducted exclusively on established, publicly available datasets intended for academic audio and bioacoustics research. Our focus on probing as an evaluation method promotes computational efficiency, significantly reducing the energy consumption and environmental impact compared to full model fine-tuning. The methods developed are for the purpose of model analysis and present no foreseeable societal risks or ethical concerns.

## Reproducibility Statement

To ensure full reproducibility, we make our source code, including the implementation of our proposed prototypical probe and all evaluation scripts, publicly available on GitHub. To further aid reproducibility and standardize access, we have also uploaded any datasets used in this study that were not not previously available on the Hugging Face Hub to the platform.

- https://github.com/lurauch/unmute-patch-tokens/
- https://huggingface.co/datasets/lrauch/desed
- https://huggingface.co/datasets/lrauch/spass
- https://huggingface.co/datasets/lrauch/urban-sed

Our experimental setup, including the specific datasets Section D.1, pretrained backbones, and pooling methods Section D.3, is detailed in Section 4.1 and in Appendix D. Appendix D.4 also provides a complete breakdown of our hyperparameter selection protocol with the respective ranges.

## Use of Large Language Models

An LLM was utilized as a writing and coding assistant during the preparation of this paper. The model was used to aid in literature discovery by summarizing concepts and identifying potentially relevant papers for the authors' review. Additionally, the LLM served as a writing aid to refine grammar, improve sentence structure, and enhance the overall clarity and readability of the paper (e.g., shorten a paragraph). It was also used for streamlining code, debugging, and generating shell scripts to help manage the experimental workflow. All research ideas, including experimental design, code implementations, and analysis of results stem from the authors without LLM involvement. The authors directed all queries, critically reviewed and carefully edited all model-generated text, and take full responsibility for the final content of this paper.

## Acknowledgements

This research was conducted under the DeepBirdDetect project (FKZ 67KI31040E), funded by the German Federal Ministry for the Environment, Nature Conservation, Nuclear Safety and Consumer

Protection (BMUV), and the BioDroneAI project (FKZ 02WDG1758D), funded by the German Federal Ministry of Research, Technology and Space (BMFTR)

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

# A  DETAILED BENCHMARK RESULTS

This appendix provides supplementary material to the benchmark evaluation presented in the main paper. Our full benchmark spans 5 general multi-label datasets, 7 few-shot bioacoustic datasets, and 2 multi-class control tasks across 6 backbones (plus 3 supervised[+] checkpoints) and 10 pooling methods. The following tables present the complete results, with all performance metrics reported as mean average precision (mAP) averaged for multi-label and accuracy for multi-class tasks over 5 random seeds.

Table 4: **Complete benchmark probing results for general multi-label audio.** This table presents the full benchmark results, extending those in the main paper with the inclusion of Supervised[+] fine-tuned checkpoints for BEATs, EAT, and SSLAM. All results are the mean mAP with standard deviation, averaged over 5 seeds. The best and second-best performing probes for each configuration are highlighted.

| Input | [cls] Baseline | | Token Map | | Token Map (Att.) | | | | Token Map (Proto.) | |
|---|---|---|---|---|---|---|---|---|---|---|
| Backbone | linear | mlp | linearc | conv | mhca | ep | simpool | abmilp | proto | protobin |
| **as20k** | | | | | | | | | | |
| A-MAE | 8.36±0.01 | 8.77±0.29 | 9.66±0.22 | 11.87±1.10 | 17.09±0.22 | 17.03±0.05 | 14.69±0.02 | 14.24±0.85 | 21.61±0.26 | 22.32±0.12 |
| ASiT | 18.35±0.01 | 19.16±0.13 | 13.36±0.12 | 13.80±0.19 | 18.72±0.17 | 18.95±0.07 | 18.04±0.01 | 16.10±0.51 | 21.89±0.06 | 20.96±0.02 |
| Dasheng | 20.98±0.06 | 21.09±0.07 | 18.23±0.11 | 18.57±1.06 | 27.49±0.07 | 26.53±0.05 | 20.89±0.01 | 22.96±1.94 | 27.59±0.07 | 29.94±0.15 |
| BEATs | 24.71±0.01 | 26.29±0.13 | 15.70±0.01 | 12.80±1.06 | 21.86±0.16 | 20.81±0.36 | 14.99±0.05 | 12.52±1.86 | 30.54±0.06 | 31.54±0.06 |
| BEATs+ | 40.30±0.02 | 40.77±0.10 | 31.33±0.15 | 34.29±0.17 | 37.57±0.13 | 37.23±0.36 | 27.38±0.08 | 30.49±2.76 | 42.73±0.06 | 41.96±0.05 |
| EAT | 17.29±0.01 | 20.59±0.16 | 21.94±0.01 | 19.50±0.34 | 26.11±0.16 | 26.83±0.04 | 25.15±0.04 | 19.91±3.40 | 31.06±0.04 | 31.67±0.06 |
| EAT+ | 44.32±0.02 | 45.31±0.06 | 37.44±0.16 | 41.85±0.15 | 41.87±0.24 | 42.53±0.09 | 41.66±0.02 | 39.14±0.42 | 43.36±0.05 | 44.64±0.02 |
| SSLAM | 17.04±0.01 | 19.99±0.08 | 20.51±0.06 | 17.45±0.54 | 24.45±0.18 | 25.49±0.01 | 22.59±0.06 | 18.91±4.42 | 30.84±0.03 | 30.94±0.08 |
| SSLAM+ | 45.72±0.02 | 46.59±0.09 | 37.61±0.02 | 43.77±0.17 | 43.40±0.07 | 44.41±0.07 | 43.37±0.06 | 41.31±0.76 | 44.64±0.06 | 43.70±0.09 |
| **fsd50k** | | | | | | | | | | |
| A-MAE | 19.71±0.03 | 21.34±0.43 | 25.17±0.74 | 40.59±0.78 | 45.17±0.45 | 43.23±0.14 | 34.89±0.05 | 32.73±4.31 | 49.65±0.17 | 49.69±0.38 |
| ASiT | 39.57±0.07 | 41.89±0.26 | 9.87±0.48 | 38.23±0.78 | 42.28±0.30 | 41.76±0.11 | 37.78±0.06 | 39.59±3.50 | 48.25±0.09 | 46.70±0.18 |
| Dasheng | 38.08±0.17 | 39.56±0.15 | 37.74±0.51 | 48.88±0.79 | 52.95±0.19 | 52.44±0.04 | 43.94±0.04 | 43.79±3.49 | 55.23±0.09 | 57.31±0.02 |
| BEATs | 46.89±0.03 | 49.58±0.31 | 36.35±0.12 | 37.19±1.63 | 48.51±0.29 | 46.16±0.07 | 40.20±0.03 | 40.32±3.22 | 57.17±0.14 | 58.27±0.15 |
| BEATs+ | 60.72±0.00 | 61.87±0.11 | 50.17±0.41 | 56.32±0.48 | 60.01±0.10 | 55.97±1.20 | 48.30±0.09 | 53.88±3.87 | 65.39±0.08 | 66.09±0.13 |
| EAT | 36.39±0.03 | 44.82±0.08 | 38.36±0.30 | 46.64±0.45 | 51.06±0.29 | 51.29±0.10 | 49.38±0.07 | 45.93±4.36 | 56.07±0.11 | 55.64±0.37 |
| EAT+ | 66.11±0.01 | 67.84±0.01 | 56.50±0.71 | 67.01±0.21 | 64.37±0.26 | 65.01±0.07 | 63.63±0.04 | 61.45±0.47 | 67.15±0.16 | 66.45±0.33 |
| SSLAM | 36.06±0.01 | 44.26±0.24 | 37.21±0.43 | 43.50±1.36 | 51.48±0.51 | 50.83±0.06 | 49.86±0.23 | 46.38±2.44 | 56.93±0.05 | 56.99±0.13 |
| SSLAM+ | 65.36±0.01 | 67.36±0.06 | 55.12±0.21 | 65.64±0.06 | 64.28±0.05 | 64.53±0.06 | 63.31±0.07 | 61.88±0.53 | 66.55±0.02 | 66.02±0.29 |
| **desed** | | | | | | | | | | |
| A-MAE | 57.46±0.01 | 60.52±0.13 | 60.88±0.14 | 84.10±0.31 | 83.57±0.20 | 80.13±0.05 | 72.05±0.03 | 76.69±0.27 | 84.11±0.07 | 85.57±0.10 |
| ASiT | 72.92±0.04 | 74.19±0.20 | 57.49±0.10 | 81.59±0.18 | 79.50±0.44 | 76.66±0.02 | 73.57±0.02 | 76.58±0.46 | 82.08±0.19 | 81.74±0.19 |
| Dasheng | 68.39±0.03 | 68.76±0.14 | 72.48±0.01 | 85.32±0.96 | 84.53±0.11 | 82.74±0.02 | 75.40±0.01 | 76.48±4.54 | 85.90±0.14 | 86.14±0.28 |
| BEATs | 77.56±0.03 | 80.56±0.15 | 72.23±0.01 | 86.83±0.55 | 86.91±0.04 | 81.88±0.04 | 81.08±0.05 | 81.77±0.95 | 89.04±0.08 | 89.22±0.55 |
| BEATs+ | 87.20±0.01 | 87.92±0.02 | 86.93±0.02 | 90.34±0.14 | 90.22±0.05 | 87.94±0.28 | 85.52±0.34 | 86.33±1.29 | 92.17±0.06 | 92.41±0.25 |
| EAT | 76.15±0.02 | 80.92±0.02 | 77.90±0.08 | 86.68±0.33 | 86.06±0.19 | 84.13±0.08 | 83.43±0.01 | 78.80±5.63 | 88.70±0.06 | 88.82±0.11 |
| EAT+ | 89.49±0.04 | 89.82±0.08 | 89.03±0.04 | 91.42±0.21 | 90.49±0.05 | 89.26±0.09 | 89.03±0.03 | 88.97±0.18 | 91.93±0.16 | 91.69±0.14 |
| SSLAM | 72.49±0.01 | 77.96±0.14 | 76.82±0.17 | 85.55±0.27 | 85.44±0.10 | 83.77±0.02 | 83.59±0.03 | 81.69±0.74 | 87.69±0.19 | 88.33±0.29 |
| SSLAM+ | 89.39±0.01 | 89.69±0.07 | 88.04±0.03 | 91.10±0.17 | 90.14±0.23 | 89.11±0.03 | 88.88±0.06 | 88.43±0.64 | 91.70±0.05 | 91.45±0.27 |
| **spass** | | | | | | | | | | |
| A-MAE | 58.94±0.03 | 60.56±0.11 | 69.01±0.66 | 80.04±0.78 | 79.24±0.14 | 71.01±0.38 | 69.84±0.02 | 68.75±0.20 | 78.92±0.24 | 79.95±0.64 |
| ASiT | 68.80±0.01 | 70.27±0.20 | 46.44±4.47 | 73.26±1.10 | 75.76±0.45 | 69.44±0.02 | 69.04±0.02 | 68.36±0.63 | 73.66±0.09 | 74.69±0.18 |
| Dasheng | 66.89±0.01 | 64.07±0.18 | 76.76±0.49 | 75.05±0.69 | 80.71±0.27 | 73.62±0.03 | 74.16±0.01 | 72.02±0.03 | 76.64±0.22 | 80.93±0.47 |
| BEATs | 74.22±0.01 | 75.97±0.14 | 79.91±0.54 | 84.81±1.49 | 83.98±0.16 | 76.61±0.09 | 75.58±0.03 | 69.38±0.41 | 87.76±0.24 | 85.77±0.43 |
| BEATs+ | 78.46±0.02 | 80.24±0.15 | 80.30±0.00 | 85.52±1.20 | 84.84±0.11 | 79.39±0.26 | 76.64±0.01 | 74.28±3.94 | 89.15±0.06 | 87.85±0.20 |
| EAT | 65.96±0.01 | 71.55±0.23 | 84.49±0.02 | 79.15±0.63 | 83.95±0.32 | 77.35±0.01 | 76.55±0.03 | 64.44±9.04 | 83.09±0.83 | 85.64±0.29 |
| EAT+ | 79.20±0.01 | 80.85±0.30 | 87.08±0.01 | 88.31±1.22 | 86.05±0.11 | 80.43±0.02 | 79.76±0.02 | 79.91±0.26 | 88.74±0.20 | 88.48±0.26 |
| SSLAM | 68.28±0.00 | 73.05±0.03 | 83.06±0.25 | 79.43±1.82 | 83.45±0.26 | 76.58±0.02 | 76.09±0.04 | 72.42±1.39 | 85.90±0.41 | 86.01±0.13 |
| SSLAM+ | 78.82±0.00 | 80.67±0.04 | 86.83±0.02 | 87.56±0.61 | 85.97±0.06 | 79.63±0.02 | 79.04±0.02 | 79.06±0.65 | 88.26±0.18 | 88.17±0.26 |
| **urban** | | | | | | | | | | |
| A-MAE | 58.72±0.06 | 58.97±0.19 | 40.53±1.18 | 85.28±0.16 | 82.49±0.16 | 79.83±0.17 | 76.21±0.07 | 73.07±2.46 | 83.63±0.19 | 85.17±0.32 |
| ASiT | 77.53±0.01 | 77.55±0.15 | 44.53±3.92 | 82.12±0.51 | 79.93±0.33 | 78.48±0.04 | 77.25±0.05 | 76.76±1.58 | 82.35±0.24 | 82.28±0.16 |
| Dasheng | 69.61±0.10 | 69.07±0.17 | 75.80±0.13 | 85.76±0.59 | 84.59±0.16 | 82.31±0.09 | 79.04±0.05 | 77.28±0.81 | 85.97±0.31 | 86.55±0.13 |
| BEATs | 82.54±0.05 | 83.76±0.04 | 75.90±0.08 | 85.57±0.48 | 86.23±0.24 | 84.31±0.12 | 82.74±0.01 | 77.89±1.07 | 89.04±0.10 | 88.74±0.15 |
| BEATs+ | 87.70±0.01 | 87.79±0.02 | 83.84±4.13 | 89.15±0.19 | 89.02±0.30 | 87.72±0.23 | 86.51±0.36 | 84.24±1.11 | 91.12±0.09 | 91.25±0.19 |
| EAT | 77.76±0.04 | 81.58±0.05 | 78.45±0.08 | 86.35±1.14 | 86.43±0.03 | 85.40±0.01 | 83.58±0.05 | 79.93±2.00 | 89.11±0.12 | 89.24±0.20 |
| EAT+ | 88.43±0.01 | 88.56±0.09 | 87.25±0.02 | 90.64±0.35 | 89.23±0.15 | 88.33±0.02 | 87.80±0.08 | 87.28±0.49 | 91.63±0.13 | 91.31±0.17 |
| SSLAM | 75.86±0.02 | 80.64±0.05 | 77.97±0.07 | 86.23±1.54 | 86.45±0.30 | 84.87±0.02 | 83.21±0.04 | 80.12±1.58 | 88.82±0.17 | 89.05±0.38 |
| SSLAM+ | 88.10±0.02 | 88.24±0.02 | 86.52±0.02 | 90.38±0.55 | 88.84±0.20 | 87.84±0.10 | 87.49±0.13 | 86.07±0.94 | 91.24±0.09 | 90.93±0.07 |

Table 5: **Benchmark probing results for few-shot multi-label bioacoustics.** This table presents the benchmark results where the figures in the main text are extracted from. All results are the mean mAP with standard deviation, averaged over 5 seeds. The **best** and second-best performing probes for each configuration are highlighted.

| | Backbone | [cls] Baseline | | Token Map (Att.) | | Token Map (Proto.) | |
|---|---|---|---|---|---|---|---|
| | | linear | mlp | mhca | ep | proto | protobin |
| hsn | A-MAE | $4.83_{\pm0.02}$ | $5.08_{\pm0.09}$ | $28.32_{\pm1.22}$ | $15.62_{\pm0.37}$ | $34.63_{\pm1.45}$ | $34.55_{\pm0.26}$ |
| | ASiT | $5.51_{\pm1.23}$ | $6.13_{\pm0.08}$ | $9.47_{\pm0.93}$ | $6.11_{\pm0.16}$ | $12.24_{\pm1.06}$ | $13.87_{\pm0.98}$ |
| | BEATs | $10.29_{\pm0.04}$ | $10.52_{\pm1.07}$ | $24.86_{\pm2.68}$ | $16.23_{\pm1.72}$ | $33.67_{\pm1.52}$ | $32.01_{\pm1.72}$ |
| | BEATs+ | $12.70_{\pm0.13}$ | $12.92_{\pm0.72}$ | $20.07_{\pm1.45}$ | $16.25_{\pm4.25}$ | $28.89_{\pm1.77}$ | $28.44_{\pm2.24}$ |
| | Dasheng | $8.05_{\pm0.19}$ | $6.65_{\pm0.09}$ | $20.78_{\pm1.37}$ | $17.56_{\pm0.45}$ | $22.04_{\pm0.66}$ | $23.43_{\pm0.61}$ |
| | EAT | $10.42_{\pm0.02}$ | $11.32_{\pm0.52}$ | $18.53_{\pm1.70}$ | $10.03_{\pm0.32}$ | $20.12_{\pm5.53}$ | $26.08_{\pm2.16}$ |
| | EAT+ | $18.43_{\pm0.46}$ | $18.86_{\pm0.55}$ | $22.43_{\pm0.86}$ | $17.61_{\pm0.16}$ | $26.42_{\pm2.81}$ | $25.97_{\pm1.14}$ |
| | SSLAM | $8.14_{\pm0.01}$ | $8.42_{\pm0.07}$ | $22.76_{\pm0.66}$ | $14.98_{\pm0.72}$ | $21.85_{\pm2.76}$ | $25.41_{\pm0.75}$ |
| | SSLAM+ | $19.31_{\pm0.68}$ | $21.05_{\pm0.48}$ | $22.83_{\pm2.10}$ | $19.86_{\pm0.13}$ | $30.00_{\pm1.50}$ | $29.85_{\pm1.20}$ |
| pow | A-MAE | $11.04_{\pm0.05}$ | $9.86_{\pm0.44}$ | $25.27_{\pm0.79}$ | $22.35_{\pm0.08}$ | $30.08_{\pm1.02}$ | $31.76_{\pm0.73}$ |
| | ASiT | $10.74_{\pm0.07}$ | $10.66_{\pm0.03}$ | $13.16_{\pm0.02}$ | $10.52_{\pm4.19}$ | $14.11_{\pm0.83}$ | $14.44_{\pm0.55}$ |
| | BEATs | $16.96_{\pm0.05}$ | $16.32_{\pm0.21}$ | $22.81_{\pm1.48}$ | $17.43_{\pm1.04}$ | $30.91_{\pm2.46}$ | $30.48_{\pm1.71}$ |
| | BEATs+ | $15.67_{\pm0.03}$ | $16.23_{\pm0.54}$ | $21.98_{\pm0.39}$ | $15.68_{\pm0.52}$ | $30.83_{\pm1.89}$ | $29.27_{\pm1.77}$ |
| | Dasheng | $13.31_{\pm0.06}$ | $12.06_{\pm0.22}$ | $17.29_{\pm0.57}$ | $15.52_{\pm0.06}$ | $19.69_{\pm0.65}$ | $19.42_{\pm0.41}$ |
| | EAT | $14.60_{\pm0.03}$ | $14.04_{\pm0.66}$ | $21.02_{\pm1.14}$ | $18.39_{\pm0.08}$ | $24.84_{\pm1.34}$ | $28.37_{\pm0.59}$ |
| | EAT+ | $17.26_{\pm0.15}$ | $19.60_{\pm0.37}$ | $23.89_{\pm2.14}$ | $18.73_{\pm0.67}$ | $31.98_{\pm0.91}$ | $30.76_{\pm1.16}$ |
| | SSLAM | $10.63_{\pm0.00}$ | $11.35_{\pm0.43}$ | $22.01_{\pm1.07}$ | $16.93_{\pm0.19}$ | $26.94_{\pm1.79}$ | $26.59_{\pm1.33}$ |
| | SSLAM+ | $16.15_{\pm0.82}$ | $17.30_{\pm1.08}$ | $23.27_{\pm0.58}$ | $16.63_{\pm0.04}$ | $27.75_{\pm1.53}$ | $28.56_{\pm2.39}$ |
| per | A-MAE | $4.01_{\pm0.01}$ | $3.78_{\pm0.20}$ | $9.66_{\pm0.37}$ | $9.50_{\pm0.07}$ | $15.48_{\pm0.36}$ | $15.00_{\pm0.70}$ |
| | ASiT | $3.31_{\pm0.02}$ | $5.76_{\pm0.44}$ | $5.60_{\pm0.33}$ | $4.94_{\pm0.04}$ | $7.38_{\pm0.17}$ | $7.57_{\pm0.20}$ |
| | BEATs | $6.00_{\pm0.03}$ | $5.74_{\pm0.38}$ | $9.58_{\pm0.07}$ | $7.72_{\pm0.24}$ | $15.16_{\pm0.16}$ | $14.40_{\pm0.27}$ |
| | BEATs+ | $6.50_{\pm0.05}$ | $6.91_{\pm0.26}$ | $10.64_{\pm0.45}$ | $7.60_{\pm0.73}$ | $14.93_{\pm0.45}$ | $14.93_{\pm0.23}$ |
| | Dasheng | $5.61_{\pm0.08}$ | $4.80_{\pm0.14}$ | $8.35_{\pm0.34}$ | $7.45_{\pm0.11}$ | $10.70_{\pm0.49}$ | $11.17_{\pm0.37}$ |
| | EAT | $4.92_{\pm0.01}$ | $4.98_{\pm0.29}$ | $8.67_{\pm0.44}$ | $8.34_{\pm0.13}$ | $12.18_{\pm0.50}$ | $12.79_{\pm0.25}$ |
| | EAT+ | $6.61_{\pm0.00}$ | $7.06_{\pm0.38}$ | $10.62_{\pm0.75}$ | $8.39_{\pm0.04}$ | $15.29_{\pm0.31}$ | $14.88_{\pm0.29}$ |
| | SSLAM | $4.68_{\pm0.02}$ | $4.54_{\pm0.19}$ | $10.31_{\pm0.41}$ | $8.57_{\pm0.05}$ | $11.97_{\pm1.13}$ | $13.18_{\pm0.06}$ |
| | SSLAM+ | $6.69_{\pm0.00}$ | $6.87_{\pm0.06}$ | $10.54_{\pm0.32}$ | $7.44_{\pm0.01}$ | $15.23_{\pm0.34}$ | $14.26_{\pm0.30}$ |
| nes | A-MAE | $3.45_{\pm0.00}$ | $3.25_{\pm0.45}$ | $18.52_{\pm0.46}$ | $16.64_{\pm0.12}$ | $25.83_{\pm0.39}$ | $25.98_{\pm0.48}$ |
| | ASiT | $3.83_{\pm0.06}$ | $4.93_{\pm0.41}$ | $6.13_{\pm0.53}$ | $5.48_{\pm0.06}$ | $9.25_{\pm0.44}$ | $9.67_{\pm0.22}$ |
| | BEATs | $9.09_{\pm0.06}$ | $9.52_{\pm0.08}$ | $16.36_{\pm0.19}$ | $11.22_{\pm0.98}$ | $26.36_{\pm0.61}$ | $25.07_{\pm0.55}$ |
| | BEATs+ | $11.43_{\pm0.08}$ | $11.85_{\pm0.09}$ | $16.91_{\pm0.50}$ | $12.07_{\pm0.62}$ | $24.54_{\pm0.17}$ | $23.47_{\pm0.33}$ |
| | Dasheng | $5.64_{\pm0.04}$ | $4.21_{\pm0.17}$ | $12.48_{\pm0.19}$ | $12.08_{\pm0.21}$ | $17.22_{\pm0.12}$ | $18.79_{\pm0.10}$ |
| | EAT | $7.88_{\pm0.00}$ | $8.78_{\pm0.06}$ | $16.77_{\pm0.50}$ | $13.79_{\pm0.07}$ | $21.18_{\pm0.59}$ | $22.03_{\pm0.58}$ |
| | EAT+ | $13.49_{\pm0.08}$ | $14.06_{\pm0.52}$ | $17.71_{\pm0.85}$ | $13.67_{\pm0.08}$ | $24.79_{\pm0.52}$ | $23.45_{\pm0.36}$ |
| | SSLAM | $5.66_{\pm0.01}$ | $6.73_{\pm0.25}$ | $17.82_{\pm0.35}$ | $14.98_{\pm0.21}$ | $21.65_{\pm0.40}$ | $21.93_{\pm0.73}$ |
| | SSLAM+ | $12.82_{\pm0.13}$ | $14.58_{\pm0.09}$ | $18.25_{\pm0.64}$ | $13.65_{\pm0.10}$ | $25.65_{\pm0.82}$ | $25.54_{\pm0.72}$ |
| sne | A-MAE | $6.09_{\pm0.04}$ | $5.90_{\pm0.12}$ | $17.56_{\pm0.06}$ | $13.48_{\pm0.05}$ | $20.23_{\pm0.91}$ | $21.38_{\pm1.39}$ |
| | ASiT | $6.41_{\pm0.13}$ | $6.92_{\pm0.46}$ | $7.12_{\pm0.41}$ | $6.79_{\pm0.11}$ | $9.63_{\pm0.32}$ | $9.63_{\pm0.35}$ |
| | BEATs | $9.61_{\pm0.05}$ | $9.95_{\pm0.10}$ | $13.91_{\pm0.80}$ | $12.16_{\pm0.29}$ | $20.26_{\pm1.26}$ | $19.46_{\pm1.56}$ |
| | BEATs+ | $11.18_{\pm0.08}$ | $12.05_{\pm0.16}$ | $12.31_{\pm0.11}$ | $10.64_{\pm0.32}$ | $17.36_{\pm0.95}$ | $16.83_{\pm0.77}$ |
| | Dasheng | $8.72_{\pm0.09}$ | $7.30_{\pm0.66}$ | $12.60_{\pm0.40}$ | $11.58_{\pm0.17}$ | $15.68_{\pm0.27}$ | $17.60_{\pm0.65}$ |
| | EAT | $10.29_{\pm0.05}$ | $10.67_{\pm0.07}$ | $14.56_{\pm0.77}$ | $11.89_{\pm0.06}$ | $16.48_{\pm0.30}$ | $16.70_{\pm0.53}$ |
| | EAT+ | $9.63_{\pm0.01}$ | $10.04_{\pm0.48}$ | $12.41_{\pm0.45}$ | $9.66_{\pm0.17}$ | $16.19_{\pm0.63}$ | $15.40_{\pm0.83}$ |
| | SSLAM | $9.52_{\pm0.05}$ | $9.96_{\pm0.02}$ | $13.42_{\pm0.58}$ | $11.04_{\pm0.06}$ | $16.25_{\pm0.57}$ | $16.40_{\pm0.29}$ |
| | SSLAM+ | $10.23_{\pm0.13}$ | $10.34_{\pm0.89}$ | $14.02_{\pm0.44}$ | $10.91_{\pm0.34}$ | $17.64_{\pm0.42}$ | $16.87_{\pm1.13}$ |
| uhh | A-MAE | $4.88_{\pm1.05}$ | $4.28_{\pm0.17}$ | $10.70_{\pm0.15}$ | $8.42_{\pm0.44}$ | $12.17_{\pm0.29}$ | $12.57_{\pm0.34}$ |
| | ASiT | $5.44_{\pm0.02}$ | $6.65_{\pm0.07}$ | $6.67_{\pm0.32}$ | $6.04_{\pm0.37}$ | $6.94_{\pm0.23}$ | $6.12_{\pm0.30}$ |
| | BEATs | $9.74_{\pm0.05}$ | $10.74_{\pm0.55}$ | $10.21_{\pm0.71}$ | $7.93_{\pm0.72}$ | $12.11_{\pm0.48}$ | $12.03_{\pm0.50}$ |
| | BEATs+ | $8.95_{\pm0.16}$ | $10.10_{\pm0.29}$ | $13.47_{\pm0.61}$ | $9.71_{\pm0.77}$ | $15.02_{\pm2.57}$ | $17.27_{\pm1.20}$ |
| | Dasheng | $5.09_{\pm0.38}$ | $5.12_{\pm0.33}$ | $7.78_{\pm0.20}$ | $6.57_{\pm0.57}$ | $7.99_{\pm0.50}$ | $10.10_{\pm0.16}$ |
| | EAT | $9.32_{\pm0.07}$ | $9.06_{\pm0.50}$ | $10.98_{\pm0.50}$ | $9.55_{\pm0.12}$ | $8.67_{\pm0.73}$ | $11.34_{\pm1.44}$ |
| | EAT+ | $10.12_{\pm0.03}$ | $10.32_{\pm0.13}$ | $10.68_{\pm0.75}$ | $9.95_{\pm0.54}$ | $13.00_{\pm0.42}$ | $13.07_{\pm0.63}$ |
| | SSLAM | $7.36_{\pm0.02}$ | $8.51_{\pm0.39}$ | $10.60_{\pm1.59}$ | $8.61_{\pm0.25}$ | $11.43_{\pm0.69}$ | $10.56_{\pm0.71}$ |
| | SSLAM+ | $10.30_{\pm0.13}$ | $11.20_{\pm0.33}$ | $9.78_{\pm0.14}$ | $9.07_{\pm0.05}$ | $13.54_{\pm0.26}$ | $12.46_{\pm0.41}$ |
| nbp | A-MAE | $9.82_{\pm0.13}$ | $8.98_{\pm0.26}$ | $32.52_{\pm0.95}$ | $26.89_{\pm0.14}$ | $40.67_{\pm0.70}$ | $41.47_{\pm1.44}$ |
| | ASiT | $12.44_{\pm0.22}$ | $13.19_{\pm0.27}$ | $16.33_{\pm0.09}$ | $14.90_{\pm0.33}$ | $19.77_{\pm0.82}$ | $20.57_{\pm0.82}$ |
| | BEATs | $17.72_{\pm0.36}$ | $19.31_{\pm0.62}$ | $28.83_{\pm1.78}$ | $17.44_{\pm0.85}$ | $41.78_{\pm1.09}$ | $40.49_{\pm1.32}$ |
| | BEATs+ | $21.41_{\pm0.43}$ | $22.60_{\pm0.30}$ | $32.58_{\pm0.74}$ | $22.92_{\pm0.73}$ | $42.84_{\pm0.73}$ | $42.10_{\pm1.10}$ |
| | Dasheng | $18.04_{\pm0.13}$ | $13.68_{\pm1.13}$ | $27.45_{\pm0.14}$ | $25.93_{\pm0.10}$ | $32.12_{\pm0.59}$ | $35.09_{\pm0.57}$ |
| | EAT | $14.59_{\pm0.11}$ | $16.95_{\pm0.51}$ | $26.86_{\pm1.17}$ | $22.14_{\pm0.33}$ | $34.39_{\pm1.07}$ | $34.40_{\pm0.59}$ |
| | EAT+ | $23.91_{\pm0.16}$ | $25.46_{\pm1.06}$ | $32.54_{\pm0.22}$ | $24.41_{\pm0.31}$ | $40.74_{\pm0.34}$ | $39.49_{\pm0.98}$ |
| | SSLAM | $10.41_{\pm0.07}$ | $14.58_{\pm0.78}$ | $27.31_{\pm0.47}$ | $20.62_{\pm0.15}$ | $34.39_{\pm1.17}$ | $34.76_{\pm0.74}$ |
| | SSLAM+ | $20.84_{\pm0.27}$ | $21.69_{\pm0.21}$ | $30.32_{\pm0.38}$ | $21.79_{\pm0.19}$ | $39.86_{\pm0.91}$ | $37.44_{\pm1.34}$ |

Table 6: **Benchmark probing results for general multi-class audio.** All results are the mean accuracy with standard deviation, averaged over 5 seeds. The best and second-best performing probes for each configuration are highlighted.

|  | Backbone | linear | mhca | protobin |
|---|---|---|---|---|
| esc50 | A–MAE | 22.08±0.14 | 86.25±0.50 | 83.67±1.42 |
|  | ASiT | 76.08±0.76 | 78.25±0.50 | 79.25±1.64 |
|  | BEATs | 78.92±0.29 | 83.17±0.76 | 84.08±1.70 |
|  | BEATs+ | 94.33±0.14 | 94.25±0.43 | 94.58±0.29 |
|  | Dasheng | 54.75±1.75 | 90.17±0.52 | 85.33±0.63 |
|  | EAT | 75.33±0.95 | 89.83±1.44 | 86.83±0.38 |
|  | EAT+ | 96.67±0.29 | 96.67±0.29 | 96.67±0.63 |
|  | SSLAM | 74.17±0.14 | 89.00±0.50 | 84.67±0.38 |
|  | SSLAM+ | 97.17±0.14 | 97.17±0.14 | 96.75±0.50 |
| ks2 | A–MAE | 12.44±1.67 | 84.87±1.19 | 79.47±1.90 |
|  | ASiT | 62.23±0.22 | 86.26±0.13 | 89.52±0.58 |
|  | BEATs | 87.00±0.18 | 94.99±0.25 | 96.53±0.11 |
|  | BEATs+ | 85.79±0.27 | 93.60±0.27 | 94.72±0.53 |
|  | Dasheng | 78.57±0.48 | 98.13±0.28 | 98.40±0.16 |
|  | EAT | 69.14±0.14 | 93.22±0.13 | 90.44±2.06 |
|  | EAT+ | 83.27±0.02 | 94.42±0.20 | 95.75±0.15 |
|  | SSLAM | 64.75±0.20 | 93.75±0.62 | 91.86±0.78 |
|  | SSLAM+ | 83.57±0.03 | 94.24±0.39 | 95.73±0.33 |

## B ABLATION STUDY

We conduct an ablation study to investigate two key aspects of our prototypical probes. First, we analyze the sensitivity to the number of prototypes ($J$) per class to justify our choice in the main benchmark. Second, we aim to disentangle the performance effects of our two main contributions: the architectural simplifications (class-agnostic design, no orthogonality loss) and the binarization itself. To achieve this, we compare three methods:

1. `proto`: The baseline from Rauch et al. (2025a) using float-based, class-dependent prototypes with an orthogonality loss.

2. `protobin`: Our proposed method using binarized, class-agnostic prototypes without an orthogonality loss.

3. `protofloat`: A new ablation variant that uses `protobin`'s simplified, class-agnostic architecture but with float-based prototypes. This allows us to isolate the impact of binarization.

The results across three diverse datasets (multi-label, high number of classes: `as20k`, multi-label low number of classes: `urban`, multi-class: `esc50`) are presented in Table 7.

**Sensitivity to number of prototypes.** Our results show a clear trend across backbones and prototypical probes: performance is highly sensitive to $J$ at lower values and begins to saturate as $J$ increases. The jump in performance from $J = 1$, $J = 5$ and $J = 10$ is notable on all datasets, though the impact varies on the task (multi-class vs. multi-label) and the dataset's structure (e.g., number of classes). For instance, on `urban` with EAT, `protobin` increases from 80.05 mAP at $J$=1 to 89.01 with nearly 9 percentage points ($\%_p$). The effect is even more pronounced on `esc50`, which sees a 14 $\%_p$ increase in the same setting. The subsequent gain from 10 to 20 is only 0.23 $\%_p$ on `urban`. The saturation suggests that while multiple prototypes are crucial, there are only diminishing returns after enough prototypes are added. In contrast, the performance difference on `as20k` is much less pronounced. Using the same EAT model, the gain from $J = 1$ to $J = 10$ is only circa 2.6 $\%_p$. This suggests that the multi-label `as20k` dataset, with its high number of classes (527), does not require as many prototypes per class, and that using a single prototype is not as detrimental

Table 7: **Comparison of probe methods across $J$ number of prototypes.** The methods include the `linear` baseline, `protobin`, `proto` and the ablation to the binarization `protofloat`. We additionally add `linear` as the baseline performance. We report the mean mAP for `as20k` and `urban`, and mean accuracy for `esc50` across 3 seeds after our hyperparameter selection. The `linear` baseline is static across $J$. **Bold** marks the number of prototypes used in our main benchmark results.

| Backbone | Probe | as20k (mAP) | | | | urban (mAP) | | | | esc50 (Accuracy) | | | |
|---|---|---|---|---|---|---|---|---|---|---|---|---|---|
| | | $J$=1 | $J$=5 | **J=10** | $J$=20 | $J$=1 | $J$=5 | $J$=10 | **J=20** | $J$=1 | $J$=5 | $J$=10 | **J=20** |
| A-MAE | protobin | 20.14 | 21.91 | 22.32 | 22.40 | 73.64 | 83.80 | 84.67 | 85.17 | 55.92 | 78.92 | 81.33 | 83.70 |
| | protofloat | 20.87 | 22.55 | 23.01 | 23.07 | 77.46 | 84.69 | 85.37 | 86.01 | 67.67 | 80.00 | 81.75 | 82.10 |
| | proto | 19.08 | 21.05 | 21.61 | 21.95 | 64.03 | 83.12 | 83.02 | 83.63 | 49.75 | 73.92 | 77.25 | 82.59 |
| | linear | | | 8.36 | | | | 58.72 | | | | 22.08 | |
| BEATs | protobin | 26.70 | 27.68 | 31.54 | 31.93 | 78.23 | 87.25 | 88.12 | 88.74 | 69.50 | 82.08 | 83.25 | 84.10 |
| | protofloat | 27.64 | 30.52 | 30.89 | 31.68 | 79.90 | 87.92 | 88.60 | 88.63 | 75.75 | 84.25 | 84.58 | 84.70 |
| | proto | 27.63 | 30.47 | 30.54 | 30.66 | 77.41 | 87.55 | 88.64 | 89.04 | 77.25 | 83.67 | 84.67 | 85.08 |
| | linear | | | 24.71 | | | | 82.54 | | | | 78.92 | |
| ASiT | protobin | 20.74 | 20.21 | 20.96 | 21.71 | 78.59 | 81.35 | 81.87 | 82.28 | 75.08 | 79.25 | 80.17 | 80.30 |
| | protofloat | 21.18 | 21.57 | 21.21 | 21.30 | 79.21 | 82.09 | 82.48 | 82.19 | 76.00 | 79.25 | 79.75 | 80.12 |
| | proto | 21.31 | 21.94 | 21.89 | 20.73 | 68.89 | 81.20 | 81.89 | 82.35 | 73.50 | 78.75 | 81.95 | 82.44 |
| | linear | | | 18.35 | | | | 77.53 | | | | 76.08 | |
| EAT | protobin | 29.08 | 31.61 | 31.67 | 32.12 | 80.05 | 88.60 | 89.01 | 89.24 | 71.00 | 84.25 | 85.00 | 86.81 |
| | protofloat | 29.11 | 31.04 | 31.19 | 31.81 | 82.00 | 88.90 | 89.08 | 89.14 | 82.58 | 87.58 | 88.25 | 89.14 |
| | proto | 28.64 | 30.65 | 31.06 | 31.30 | 72.34 | 87.65 | 88.84 | 89.11 | 55.33 | 78.08 | 82.58 | 85.91 |
| | linear | | | 17.29 | | | | 77.76 | | | | 75.33 | |
| SSLAM | protobin | 28.69 | 29.77 | 30.94 | 32.10 | 81.08 | 86.82 | 88.92 | 89.05 | 65.75 | 80.67 | 83.83 | 84.70 |
| | protofloat | 29.08 | 30.50 | 30.55 | 31.26 | 81.63 | 87.16 | 89.05 | 89.05 | 79.75 | 86.00 | 86.17 | 86.69 |
| | proto | 29.08 | 30.53 | 30.84 | 30.99 | 81.60 | 88.58 | 88.45 | 88.82 | 62.08 | 80.67 | 82.17 | 85.18 |
| | linear | | | 17.04 | | | | 75.86 | | | | 74.17 | |

as it is for the multi-class tasks. Regardless of the task, this analysis confirms that our choice of $J = 20$ (and $J = 10$ for `as20k`) for the central benchmark is robust, capturing the vast majority of the method's potential performance without adding excessive parameters.

**Binarization and architectural simplification.** This ablation reveals that our architectural simplifications are the primary driver of performance gains, while binarization offers a highly effective trade-off between a very minor precision cost in certain cases and major efficiency benefits.

- *Impact of binarization* (`protobin` vs. `protofloat`): Comparing our proposed method `protobin` to its float-based counterpart `protofloat` reveals the direct impact of binarization. On some configurations in lower and higher number of prototypes, `protofloat` performs slightly better than `protobin`. The performance differences are expected and highlights an inherent trade-off: the full precision of 32-bit floats can capture finer-grained details. However, `protobin` remains highly competitive, demonstrating that binarization achieves a 32x memory reduction at the cost of only a very low drop in performance in certain cases.

- *Impact of simplification* (`protofloat` vs. `proto`): This comparison provides the cleanest evidence for the impact of our architectural changes. Our simplified, class-agnostic `protofloat` consistently and significantly outperforms the class-dependent `proto` baseline across nearly all settings. This confirms that decoupling prototypes from classes leads to better performance.

- *Overall* (`protobin` vs. `proto` vs. `linear`): We observe that `protobin` frequently outperforms the original `proto` baseline and `linear`, especially on the complex multi-label datasets `as20k` and `urban`. It shows that the benefits of our architectural simplifications (the class-agnostic design) are powerful enough to often outweigh the minor precision loss from binarization, resulting in a performance gain with a simpler and more efficient model.

**Task and dataset characteristics.** The ablation results also underscore the task-dependent nature of the different prototypical architectures.

- *Multi-label* (`as20k`, `urban`): On these complex datasets, the results underscore the advantage of our simplified and class-agnostic architecture. The presence of polyphony requires a flexible design where prototypes can collaborate to disentangle overlapping sound events, a strength of `protobin` and `protofloat`. This is particularly evident for models with highly entangled embeddings (e.g., A-MAE) and on datasets with many classes (`as20k`), where the scalability of reusable prototypes for different classes is beneficial.
- *Muti-class* (`esc50`): Conversely, on this single-label task, the advantage of our class-agnostic design diminishes. With only a single dominant sound source, the simpler and more direct supervisory signal of a class-dependent mapping can be more effective. In cases with less discriminative embeddings (e.g., ASiT), the full float precision of `proto` may also be necessary to capture fine-grained acoustic details, making it more competitive than our regularized `protobin`.

On the complex, multi-label datasets (`as20k`, `urban`), the architectural flexibility of our class-agnostic `protofloat` and `protobin` provides an advantage over the more class-dependent `proto`. This supports our core hypothesis that a disentangled design is valuable for polyphonic scenes. Conversely, on the single-label `esc50` task, this advantage diminishes. Here, the baseline `proto` is highly competitive, as the simpler challenge of learning a direct class-to-prototype mapping seems to be sufficient for single-source audio.

## C    COMPUTATIONAL RESOURCES

To motivate the upper bound calculation, our benchmark combined 14 datasets, 9 backbones, and 10 pooling methods. Each of these combinations involved up to 50 hyperparameter trials plus 5 final evaluation runs, establishing the basis for our total run count. The computational cost of our benchmark can be divided into two stages. The first was a one-time pre-computation of embeddings for each of the 9 backbone checkpoints across all 14 datasets. For the 7 general audio and control datasets, we generated embeddings once for each of the 9 backbones. For the 7 bioacoustic datasets, this process was repeated 5 times per backbone to create distinct augmented variants for training. This initial stage resulted in:

$$\underbrace{9}_{\text{backbones}} \cdot \left( \underbrace{7}_{\text{datasets}} \cdot \underbrace{1}_{\text{run/data}} + \underbrace{7}_{\text{bio-data}} \cdot \underbrace{5}_{\text{runs/data}} \right) = 378 \text{ pre-computation runs} \qquad (4)$$

The second stage was the training and evaluation of the probing methods, where the hyperparameter optimization involved 50 initial trials managed by a successive-halving scheduler, followed by 5 final evaluation runs.

The number of pooling methods evaluated varied by dataset category. For the 5 general multi-label audio datasets, where all 10 pooling methods were evaluated, the upper bound on training runs was:

$$\underbrace{5}_{\text{datasets}} \cdot \underbrace{9}_{\text{backbones}} \cdot \underbrace{10}_{\text{probes}} \cdot \left( \underbrace{50}_{\text{HPS}} + \underbrace{5}_{\text{final seeds}} \right) = 24,750 \qquad (5)$$

For the 7 few-shot bioacoustic datasets, we used a reduced set of 6 relevant pooling methods, resulting in:

$$\underbrace{7}_{\text{datasets}} \cdot \underbrace{9}_{\text{backbones}} \cdot \underbrace{6}_{\text{probes}} \cdot \left( \underbrace{50}_{\text{HPS}} + \underbrace{5}_{\text{final seeds}} \right) = 20,790 \qquad (6)$$

Finally, for the 2 multi-class control datasets, we evaluated a core set of 3 representative probes:

$$\underbrace{2}_{\text{datasets}} \cdot \underbrace{9}_{\text{backbones}} \cdot \underbrace{3}_{\text{probes}} \cdot \left( \underbrace{50}_{\text{HPS}} + \underbrace{5}_{\text{final seeds}} \right) = 2,970 \qquad (7)$$

Summing these values gives the total upper bound on individual training runs for the entire benchmark:

$$24,750 + 20,790 + 2,970 = \mathbf{48{,}510} \text{ total runs} \tag{8}$$

We executed all benchmark tasks on a high-performance compute cluster equipped with NVIDIA A100 GPUs. This includes the initial augmentation-free forward pass required to pre-compute and cache the embeddings for all backbones, as well as the subsequent training and evaluation of all probing methods. The resulting on-disk embedding store for all cached features occupied approximately 3.6 TB of storage. Code development and preliminary tests were performed on a workstation using an NVIDIA RTX4090 GPU and an AMD Ryzen 9 7950X CPU.

## D    BENCHMARK IMPLEMENTATION DETAILS

This Appendix provides further details on the core components of our benchmark's experimental setup.

### D.1    BENCHMARK DATASETS

Table 8 presents an overview of all 14 downstream datasets used in our benchmark, categorized into three thematic groups along with their respective sizes.

Table 8: **Overview of the benchmark datasets.** The datasets are organized into three groups: general multi-label, few-shot bioacoustic multi-label, and general multi-class. For each dataset, we report the size of the train, validation, and test splits, the number of classes, and the audio clip length. Note that all bioacoustic tasks follow a 64-shot training protocol.

| Dataset | Train | Validation | Test | #Classes | Clip Length [s] |
|---|---|---|---|---|---|
| *Multi-label: General Datasets* | | | | | |
| as20k (Gemmeke et al., 2017) | 18,685 | – | 17,142 | 527 | 10 |
| desed (Johnson et al., 2021) | 20,000 | – | 2,000 | 10 | 10 |
| fsd50k (Fonseca et al., 2022) | 40,966 | – | 10,231 | 200 | 10 |
| spass (Viveros-Muñoz et al., 2023) | 17,500 | 3,750 | 3,750 | 28 | 10 |
| urban (Salamon et al., 2017) | 6,000 | 2,000 | 2,000 | 10 | 10 |
| *Multi-label: Bioacoustic BirdSet (64-shot)* | | | | | |
| hsn (Rauch et al., 2025b) | 1,344 | — | 12,000 | 21 | 5 |
| pow (Rauch et al., 2025b) | 3,072 | – | 4,560 | 48 | 5 |
| per (Rauch et al., 2025b) | 8,448 | – | 15,120 | 132 | 5 |
| nes (Rauch et al., 2025b) | 5,696 | – | 24,480 | 89 | 5 |
| sne (Rauch et al., 2025b) | 3,584 | – | 23,756 | 56 | 5 |
| uhh (Rauch et al., 2025b) | 1,600 | – | 36,367 | 27 | 5 |
| nbp (Rauch et al., 2025b) | 3,264 | – | 563 | 51 | 5 |
| *Multi-class: General Datasets* | | | | | |
| esc50 (Piczak, 2015)[‡] | 1,600 | – | 400 | 50 | 5 |
| sc2 (Warden, 2018) | 84,848 | 9,982 | 4,890 | 12 | 10 |

**AudioSet** (Gemmeke et al., 2017). `as2m` is a large-scale dataset used to pretrain general-purpose audio models and built from a vast collection of YouTube videos. It features a comprehensive ontology of over 500 sound classes, making it a standard benchmark for general-purpose audio event detection and classification. The `as20k` dataset represents a commonly used subset with 20,000 samples.

**Domestic Environment Sound Event Detection** (Johnson et al., 2021). `desed` is designed for evaluating sound event detection in domestic settings, featuring 10-second audio clips. These recordings are annotated with temporal labels for 10 common sound classes like dishes, speech, and vacuum cleaners. It was specifically created to facilitate research in both centralized and federated learning scenarios.

**Free Sound Dataset 50k** (Fonseca et al., 2022). `fsd50k` is a large, open dataset for sound event research, containing over 51,000 audio clips from the Freesound platform. It covers 200 diverse sound classes drawn from the AudioSet Ontology, with a focus on label quality through a multi-step human verification process. The dataset is widely used for multi-label sound classification and detection tasks.

**Synthetic Polyphonic Dataset with Spatiotemporal Labels of Sound Sources** (Viveros-Muñoz et al., 2023). `spass` is a synthetic collection of polyphonic soundscapes created for sound source localization and separation tasks for 28 urban sounds. It provides detailed spatiotemporal annotations, specifying the precise time, location, and class of each sound event within the clips. It contains a set of five distinct acoustic background scenes. This makes it particularly valuable for developing and testing models that can understand complex acoustic scenes.

**Urban-SED** (Salamon et al., 2017). `urban` is a collection for urban sound classification, containing 10-second audio clips of 10 common urban sound classes. These classes include events such as car horns, sirens, and street music, recorded from real-world city environments. The dataset serves as a popular benchmark for models tasked with environmental sound analysis.

**BirdSet** (Rauch et al., 2025b). BirdSet is a comprehensive, large-scale collection of datasets for avian bioacoustics research. It aggregates recordings from various global locations, with each location forming a distinct subset (`hsn`, `pow`, `per`, `nes`, `sne`, `uhh`, `nbp`). The collection is specifically tailored to benchmark audio classification models, reflecting realistic bioacoustic monitoring challenges.

**Environmental Sound Classification** (Piczak, 2015). `esc50` is a benchmark collection for Environmental Sound Classification, consisting of 2,000 five-second audio clips. It is uniformly organized into 50 distinct semantic classes, including animal sounds, natural soundscapes, and human non-speech sounds. The dataset is standardized with a pre-defined 5-fold cross-validation setup, making it a standard for evaluating audio SSL models.

**Speech Commands V2** (Warden, 2018). `sc2` is designed for keyword spotting and limited-vocabulary speech recognition. It contains thousands of one-second utterances of short command words (e.g., "up," "down," "stop") spoken by many different individuals. It contains 35 commands in the vocabulary, providing a robust benchmark for testing general-purpose models in audio.

## D.2 FEWSHOT BIRDSET DETAILS

For our few-shot learning evaluation on the seven BirdSet downstream tasks, we constructed 64-shot training subsets. The creation of these subsets follows the pipeline detailed in (Rauch et al., 2025a), which involves a selection of audio clips to mitigate label noise from weakly-labeled recordings. Full details of the subset creation process and dataset characteristics can be found in the original BirdSet publications (Rauch et al., 2025b;a). Given the challenging nature of these tasks—which are multi-class during training but multi-label during testing, we introduced a light data augmentation strategy. For each of the seven 64-shot datasets, we pre-generated and saved five distinct augmented variants using only the mixup augmentation with $p = 0.9$, which is highly effective for bird sounds (Rauch et al., 2025a). During each experimental run, we randomly selected a sample of one of these five variants for training, providing diversity to the learning process without on-the-fly computational overhead.

## D.3 POOLING METHODS

Table 9 summarizes the ten distinct pooling methods evaluated in our study. It details their architectural family, whether they operate on the `[cls]`-token or the full token map, and their computational complexity.

Table 9: **Pooling methods overview.** Methods are grouped by architectural family. The #params row lists symbolic counts, and the `urban` row instantiates them for our EAT-B/768 setup on `urban`. Symbols: $N$ tokens ($= S_t \cdot S_f$), $D$ embed dim, $C$ classes, $H$ MLP hidden, $k$ conv kernel, $D_h$ conv hidden, $F$ number of frequency patches, $Q$ queries, $J$ prototypes.

| | [cls] Baseline | | Token Map | | | Token Map (Att.) | | | | Token Map (Proto.) | |
|---|---|---|---|---|---|---|---|---|---|---|---|
| | linear | mlp | linearc | conv | linpre | mhca | ep | simpool | abmilp | proto | protobin |
| # params | $DC$ | $DH + HC$ | $NDC$ | $k^2DD_h + D_hC$ | $FDC$ | $D^2+DC$ | $D^2+QD+DC$ | $D^2+DC$ | $2D^2+QD+DC$ | $J+JC$ | $J+JC$ |
| urban | $\approx 7.7$k | $\approx 398$k | $\approx 3.9$M | $\approx 2.7$M | $\approx 61.4$k | $\approx 1.2$M | $\approx 622$k | $\approx 598$k | $\approx 3.0$M | $\approx 155$k | $\approx 155$k |

### D.4 HYPERPARAMETER SETTINGS

For each unique combination of a backbone, dataset, and pooling method, we conducted a systematic hyperparameter search to find the optimal learning rate and weight decay. This process ensures that each method is evaluated under its best-performing configuration, providing a fair comparison. Our search strategy consists of 50 trials for each combination, managed by a successive-halving pruner to improve efficiency. The search is structured in two stages. First, we explore with 25 trials, using a Sobol sequence to perform a quasi-random search, ensuring a broad and uniform coverage of the hyperparameter space. Second, we exploit with 25 trials using a tree-structured parzen estimator (TPE) to focus the search on promising regions identified during the exploration phase. The configuration yielding the highest mean Average Precision (mAP) on the validation set is then selected for the final evaluation, where it is re-trained and tested using five different random seeds.

The search spaces were kept compact. For all baseline, convolutional, and attentive pooling methods, the search space was:

- **Learning Rate (lr)**: A log-uniform distribution between $1 \times 10^{-4}$ and $7 \times 10^{-3}$.
- **Weight Decay (wd)**: A log-uniform distribution between $1 \times 10^{-5}$ and $5 \times 10^{-4}$.

Based on preliminary experiments showing that prototypical methods benefit from a higher learning rate, their search space for the learning rate was adjusted, while the weight decay remained the same:

- **Learning Rate (lr)**: A log-uniform distribution between $2 \times 10^{-3}$ and $8 \times 10^{-2}$.

All other hyperparameters were held constant across all experiments to isolate the effects of the pooling method. These fixed settings are summarized in Table 10.

Table 10: Fixed hyperparameters used for training all probing heads.

| Optimizer | Epochs | Batch Size | LR Scheduler | Loss Function | Prototypes/Class | Prototype LR |
|-----------|--------|------------|--------------|---------------|------------------|--------------|
| AdamW | 30 | 128 | Cosine | Asymmetric | 20 | Global LR |

