# OpenReview forum: "Unmute the Patch Tokens: Rethinking Probing in Multi-Label Audio Classification"
_ICLR.cc/2026/Conference — ICLR 2026 Poster_

### Official Review · Reviewer_HEBB · 2025-10-18

**Soundness:** 3
**Presentation:** 4
**Contribution:** 3
**Rating:** 8
**Confidence:** 3

**Summary:**

This paper investigates the underperformance and underutilization of probing as an evaluation paradigm for self-supervised audio models, particularly in multi-label classification tasks. The authors hypothesize that the standard approach of using a linear probe on a globally pooled [cls] token creates an information bottleneck, discarding crucial information about localized sound events contained within the patch tokens. To address this, they introduce "binarized prototypical probes" (protobin), a lightweight and efficient pooling method that learns a set of per-class prototypes to aggregate information directly from the full token map. Through a comprehensive benchmark, the authors demonstrate that their proposed method significantly outperforms standard linear and attentive probing. The work makes a strong case for establishing prototypical probing as a more faithful and efficient evaluation standard in the audio SSL community.

**Strengths:**

- Systematic and Insightful Problem Analysis: The paper's greatest strength is its methodical approach. Rather than just presenting a new method, it first provides a deep and convincing analysis of the existing problem—the "pooling bottleneck"—and validates this hypothesis with extensive empirical evidence.
- Methodological Elegance and Efficiency: The proposed protobin method is simple to understand and implement, yet highly effective. It is parameter-efficient and, through binarization, memory-efficient, making it a practical tool for a wide range of applications, including on-device scenarios.
- Comprehensive and Rigorous Benchmarking: The scale of the experimental study is a major strength. The evaluation across 13 datasets, 6 encoders (and their supervised variants), and 10 pooling methods provides a robust and generalizable foundation for the paper's claims. The structured presentation of results in response to explicit research questions is exemplary.
- Clarity of Presentation: The writing is clear, and the high-quality visualizations (especially Figures 1, 2, and 5) are instrumental in conveying the paper's core ideas and findings effectively.

**Weaknesses:**

- Fixed Number of Prototypes: The number of prototypes is set to 20 per class across all datasets, following a heuristic from a prior work. While this seems to work well, the paper would be slightly stronger with a brief sensitivity analysis or discussion on how this hyperparameter might be optimally chosen for datasets with different characteristics (e.g., number of classes, intra-class variance).
- Trade-off of Binarization: The protobin method is compared to a float-based proto version, and while it performs competitively and often wins, it is not uniformly superior. A brief discussion on the specific trade-offs of binarization (e.g., in which scenarios might the precision of float-based prototypes be advantageous?) could add more nuance.

**Questions:**

Please refer to the weaknesses above for the questions.

---

> ### Author Response · Authors · 2025-11-17
> **The Authors' Response (1)**
>
> Dear **[HEBB]**,
>
> First of all, we want to thank you for taking the time to provide an insightful review. We are particularly grateful that you recognized many of the core strengths of our work, including our systematic analysis of the pooling bottleneck (Strength 1), the methodological elegance and efficiency of protobin (Strength 2), our comprehensive benchmark (Strength 3), and the clarity of the presentation (Strength 4). Your suggestions were very helpful for strengthening the paper further. We address each point below, summarizing our response and the concrete actions taken. All changes based on your feedback are marked in [GREEN] in the revised paper.
>
> ### **W1: Prototype ablations**
> > *Fixed Number of Prototypes: The number of prototypes is set to 20 per class across all datasets, following a heuristic from a prior work. While this seems to work well, the paper would be slightly stronger with a brief sensitivity analysis or discussion on how this hyperparameter might be optimally chosen for datasets with different characteristics (e.g., number of classes, intra-class variance).*
>
> **Our reasoning and statement.**  This is a valuable suggestion. For the main benchmark, we fixed the number of prototypes to ensure a fair and controlled comparison between pooling methods. However, we agree that a sensitivity analysis adds notable value and have now included a detailed ablation study to investigate this.
>
> - *Justification for fixed J:* Our primary goal was to isolate the performance of different pooling methods. Introducing a tunable J for every combination of dataset, backbone, and probe would have added a confounding variable and made direct comparisons less reliable (and added more parameters). We therefore used a constant, reasonable value based on prior work.
> - *Intuition*: We share your intuition that the optimal number of prototypes can be task-dependent. For a real-world application, J could be tuned based on factors like:
>     - Intra-class variance: Classes with high acoustic diversity may benefit from more prototypes. A class with low variance will be well-represented by fewer prototypes.
>     - Dataset complexity: Highly polyphonic scenes may require more prototypes to effectively disentangle overlapping sounds.
>     - Task dependence: The task (e.g. multi-class vs. multi-label) could influence the required number of prototypes.
>
> **Our actions based on your feedback.**
>
> - [NEW EXPERIMENT] We have added a detailed ablation study investigating the impact of varying the number of prototypes on three diverse datasets (Appendix B: Ablation Study, L.894-983, Table 7).
> - We observe that the performance is sensitive to the number of prototypes $J$ at low values but saturates quickly as $J$ increases.
> - We also see that multi-label on as20k with over 500 classes has a much less pronounced gain from 1 to multiple prototypes per class, suggesting it does not require as many prototypes. For urban and the multi-class esc50, 1 prototype is highly detrimental.

---

> ### Author Response · Authors · 2025-11-17
> **The Authors' Response (2)**
>
> In this comment, we provide the new ablation study of Appendix B:
>
> | Backbone | Probe | as20k (J=1) | as20k (J=5) | as20k (**J=10**) | as20k (J=20) | urban (J=1) | urban (J=5) | urban (J=10) | urban (**J=20**) | esc50 (J=1) | esc50 (J=5) | esc50 (J=10) | esc50 (**J=20**) |
> |:---|:---|:--:|:--:|:--:|:--:|:--:|:--:|:--:|:--:|:--:|:--:|:--:|:--:|
> | A-MAE | protobin | 20.14 | 21.91 | 22.32 | 22.40 | 73.64 | 83.80 | 84.67 | 85.17 | 55.92 | 78.92 | 81.33 | 83.70 |
> | A-MAE | protofloat | 20.87 | 22.55 | 23.01 | 23.07 | 77.46 | 84.69 | 85.37 | 86.01 | 67.67 | 80.00 | 81.75 | 82.10 |
> | A-MAE | proto | 19.08 | 21.05 | 21.61 | 21.95 | 64.03 | 83.12 | 83.02 | 83.63 | 49.75 | 73.92 | 77.25 | 82.59 |
> | A-MAE | linear | 8.36 | 8.36 | 8.36 | 8.36 | 58.72 | 58.72 | 58.72 | 58.72 | 22.08 | 22.08 | 22.08 | 22.08 |
> | BEATs | protobin | 26.70 | 27.68 | 31.54 | 31.93 | 78.23 | 87.25 | 88.12 | 88.74 | 69.50 | 82.08 | 83.25 | 84.10 |
> | BEATs | protofloat | 27.64 | 30.52 | 30.89 | 31.68 | 79.90 | 87.92 | 88.60 | 88.63 | 75.75 | 84.25 | 84.58 | 84.70 |
> | BEATs | proto | 27.63 | 30.47 | 30.54 | 30.66 | 77.41 | 87.55 | 88.64 | 89.04 | 77.25 | 83.67 | 84.67 | 85.08 |
> | BEATs | linear | 24.71 | 24.71 | 24.71 | 24.71 | 82.54 | 82.54 | 82.54 | 82.54 | 78.92 | 78.92 | 78.92 | 78.92 |
> | ASiT | protobin | 20.74 | 20.21 | 20.96 | 21.71 | 78.59 | 81.35 | 81.87 | 82.28 | 75.08 | 79.25 | 80.17 | 80.30 |
> | ASiT | protofloat | 21.18 | 21.57 | 21.21 | 21.30 | 79.21 | 82.09 | 82.48 | 82.19 | 76.00 | 79.25 | 79.75 | 80.12 |
> | ASiT | proto | 21.31 | 21.94 | 21.89 | 20.73 | 68.89 | 81.20 | 81.89 | 82.35 | 73.50 | 78.75 | 81.95 | 82.44 |
> | ASiT | linear | 18.35 | 18.35 | 18.35 | 18.35 | 77.53 | 77.53 | 77.53 | 77.53 | 76.08 | 76.08 | 76.08 | 76.08 |
> | EAT | protobin | 29.08 | 31.61 | 31.67 | 32.12 | 80.05 | 88.60 | 89.01 | 89.24 | 71.00 | 84.25 | 85.00 | 86.81 |
> | EAT | protofloat | 29.11 | 31.04 | 31.19 | 31.81 | 82.00 | 88.90 | 89.08 | 89.14 | 82.58 | 87.58 | 88.25 | 89.14 |
> | EAT | proto | 28.64 | 30.65 | 31.06 | 31.30 | 72.34 | 87.65 | 88.84 | 89.11 | 55.33 | 78.08 | 82.58 | 85.91 |
> | EAT | linear | 17.29 | 17.29 | 17.29 | 17.29 | 77.76 | 77.76 | 77.76 | 77.76 | 75.33 | 75.33 | 75.33 | 75.33 |
> | SSLAM | protobin | 28.69 | 29.77 | 30.94 | 32.10 | 81.08 | 86.82 | 88.92 | 89.05 | 65.75 | 80.67 | 83.83 | 84.70 |
> | SSLAM | protofloat | 29.08 | 30.50 | 30.55 | 31.26 | 81.63 | 87.16 | 89.05 | 89.05 | 79.75 | 86.00 | 86.17 | 86.69 |
> | SSLAM | proto | 29.08 | 30.53 | 30.84 | 30.99 | 81.60 | 88.58 | 88.45 | 88.82 | 62.08 | 80.67 | 82.17 | 85.18 |
> | SSLAM | linear | 17.04 | 17.04 | 17.04 | 17.04 | 75.86 | 75.86 | 75.86 | 75.86 | 74.17 | 74.17 | 74.17 | 74.17 |

---

> ### Author Response · Authors · 2025-11-17
> **The Authors' Response (3/3)**
>
> ### **W2: Trade-off binarization**
>
> > *Trade-off of Binarization: The protobin method is compared to a float-based proto version, and while it performs competitively and often wins, it is not uniformly superior. A brief discussion on the specific trade-offs of binarization (e.g., in which scenarios might the precision of float-based prototypes be advantageous?) could add more nuance.*
>
> **Our reasoning and statement.** This is another excellent point. The fact that binarization is not uniformly superior is an important finding. We have revised the paper to explicitly discuss this trade-off between the precision of float prototypes and the efficiency, regularization, and architectural simplicity of protobin (also in regards to reviewer [gW4z]).  Protobin was primarily designed to create a simpler, more efficient, and elegant probing method. One of the main differences to the base proto is that we remove the need for an orthogonality loss term as well as making the prototypes class-agnostic. This is our main difference that enables a competitive (or even better) performance. The trade-off is between the precision of 32-bit floats and the efficiency gained from binarization.
>
> - When float-based proto might excel: In settings with very low inter-class variance (or hard to disentangle embeddings), where classes are extremely similar, the full precision of 32-bit floats might be necessary to capture subtle, fine-grained distinctions in the embedding space.
> - When binarization is better: The binarization can act as a regularizer, preventing overfitting and enabling robust disentanglement of features in complex scenes (as seen in Figure 1). The architectural simplifications are a primary driver of its strong performance.
>
> **Our action based on your feedback.**
>
> - [NEW EXPERIMENT] In our new ablation study mentioned above, we also include a float-proto variant that uses protobin's class-agnostic design. This provides a more direct comparison to isolate the effect of binarization itself (Appendix B: Ablation Study, L.894-983, Table 7). See markdown table above.
> - We see that protofloat’s full 32-bit precision can capture finer-grained acoustic details, which gives it a slight performance edge in some configurations (e.g., in the urban dataset with more polyhonic sounds).
> - We have added a discussion of this trade-off from proto to protobin to the Q1 results section (L.376-381).
>
> -------
>
> ### **Conclusion**
>
> **We are confident that these revisions, based on your constructive feedback, have made our paper even stronger. Thank you again for helping us improve the work. If there are any remaining weaknesses, please let us know.**

---

> > ### Comment · Reviewer_HEBB · 2025-11-24
> > **Response to rebuttal**
> >
> > I would like to thank the authors for their extensive rebuttal. The clarifications provided were very helpful in resolving my initial queries. I will be maintaining my recommendation and hope to see this work presented at the conference.

---

> > > ### Author Response · Authors · 2025-11-24
> > > **Thank you for the continued support**
> > >
> > > Thank you again for the positive feedback and your confirmation. We are glad that our rebuttal successfully clarified your initial queries and we appreciate your continued support for this work.

---

### Official Review · Reviewer_FqWa · 2025-10-29

**Soundness:** 2
**Presentation:** 3
**Contribution:** 2
**Rating:** 2
**Confidence:** 5

**Summary:**

## Rethinking probing for Audio SSL models

1. The paper proposes prototypical probing strategies for probing frozen Audio SSL models.
2. Several pooling approaches have been evaluated across several audio SSL models with different pretext tasks for a holistic evaluation of the proposed pooling approaches.
3. The empirical evaluation conducted in the paper is solid, and showcases that the proposed pooling approach works better than alternatives for pooling.

**Strengths:**

1. For the most part, the paper is well written and well presented. Diagrams and illustrations aid in understanding the paper.
2. The proposed binarised prototypical probing methods are well presented.
3. The evaluation protocol is thorough, with a good selection of audio SSL models and several probing strategies.
4. Results show that protobin probing works well, especially on 3 of the 5 evaluated tasks.

**Weaknesses:**

## Weakness 1

a. The core argument/motivating principle, repeated several times in the paper, is that "audio SSL works default to fine-tuning instead of probing", but only a handful of papers that support the argument are cited throughout the paper. E.g.

Lines 52-53
> Given probing’s apparent practicality and widespread adoption as an evaluation paradigm in computer vision (Oquab et al., 2024; Przewi˛e´zlikowski et al., 2025; Darcet et al., 2025), why does the audio SSL community still default to resource-intensive fine-tuning (Alex et al., 2025; Chen et al., 2024)?

Lines 72-73
> Therefore, the limited adoption of probing in audio SSL ....

Lines 214-215
> Evaluation in audio SSL. While probes are widely used in computer vision (Oquab et al., 2024), audio SSL defaults to fine-tuning (Rauch et al., 2025b).

Statements such as the above make it seem like the audio SSL domain is not doing probing-based evaluation on frozen encoders at all.  However, the paper only cites a handful of papers that support this notion, while ignoring the plethora of audio SSL works that evaluate frozen encoders using probing [1-15] (to mention a few). This includes Dasheng [8], the model you do cite in the paper and conduct experiments on.

---

## Weakness 2

1. Further, the core motivation/hypothesis/guiding principle of the paper is that there is a "pooling bottleneck", and that standard single vector probes and [cls]-token only pooling under utilize token embeddings. However, several missing references listed in W1 are already aware of the drawbacks of conducting probing experiments only on [cls]-token or single vector representations obtained by pooling of information across all the patches.

2. In MIM style SSL from spectrograms, every patch is a fragment in time and frequency. Thus, aggregation strategies that merge information across patches will inherently decimate frequency information. Niizumi et al. [02] were aware of this and proposed a simple strategy to aggregate temporal features while preserving frequency information.

3. Thus, the <[cls]-linear probe> strategy is not representative of the entire audio SSL community's approach to evaluate audio SSL models, unlike what the paper in its current form makes it seem like.

---

## Weakness 3

From lines 135-136
> If the model exposes a last-layer....... If not, we can mean pool $\tilde{z}_i$ by averaging all token positions.

As mentioned in Weakness 2, Point 2. This pooling strategy across token positions is suboptimal and puts plain linear probe at a disadvantage for the models that do not expose cls tokens.

---

## Weakness 4

In several instances, the paper makes unjustified statements.

> All current spectrogram-based audio SSL encoders apply MIM-style objectives, often coupled with student-teacher distillation (Chen et al., 2024; Alex et al., 2025).

These two papers are not representative of the entire audio SSL field. For e.g. [9, 13, 15] do not use a straightforward MIM style objective for learning audio representations through SSL.

> While probes are widely used in computer vision (Oquab et al., 2024), audio SSL defaults to fine-tuning (Rauch et al., 2025b).

How does the paper on the BIRDNET dataset (Rauch et al., 2025b) justify that audio SSL defaults to using fine-tuning?

---

## REFERENCES

01. Koutini et al., "Learning General Audio Representations With Large-Scale Training of Patchout Audio Transformers", 2022.
02. Niizumi et al., "Masked spectrogram modeling using masked autoencoders for learning general-purpose audio representation",   2022.
03. Anton et al., "AUDIO BARLOW TWINS: SELF-SUPERVISED AUDIO REPRESENTATION LEARNING", 2023.
04. Niizumi et al., "BYOL for Audio: Exploring Pre-Trained General-Purpose Audio Representations", 2023.
05. Niizumi et al., "Masked Modeling Duo: Towards a Universal Audio Pre-Training Framework", 2024.
06. Yadav et al., "Masked Autoencoders with Multi-Window Local-Global Attention Are Better Audio Learners", 2024.
07. Yadav et al., "Audio Mamba: Selective State Spaces for Self-Supervised Audio Representations", 2024.
08. Dinkel et al., "Scaling up masked audio encoder learning for general audio classification", 2024.
09. Li et al., "Self-Supervised Audio Teacher-Student Transformer for Both Clip-Level and Frame-Level Tasks", 2024.
10. Yadav et al., "AxLSTMs: learning self-supervised audio representations with xLSTMs", 2025.
11. Yuksel et al., "GRAM: Spatial general-purpose audio representation models for real-world applications", 2025.
12. Schmid et al., "Effective Pre-Training of Audio Transformers for Sound Event Detection", 2025.
13. Pepino et al., "EnCodecMAE: leveraging neural codecs for universal audio representation learning", 2025.
14. Niizumi et al., "M2D-CLAP: Exploring General-Purpose Audio-Language Representations Beyond CLAP", 2025.
15. Chang et al., "USAD: Universal Speech and Audio Representation via Distillation", 2025

**Questions:**

No specific questions. Please address the weaknesses stated above.

---

> ### Author Response · Authors · 2025-11-17
> **Authors' Response (1)**
>
> Dear **[FqWa]**,
>
> First of all, thank you for taking your time for this thoughtful feedback. We appreciate that you recognized our paper’s key contributions, including our novel perspective on frozen embeddings, our extensive benchmark, and the superior performance of our proposed approach  over other probing techniques. You correctly identified that the paper's positioning can be improved. Your comments have been very valuable in helping us better clarify our work's context and contribution. Below, we detail how we have addressed this positioning weakness and all your other concerns. We have gone through your mentioned works and very carefully added changes in the manuscript to improve it based on your feedback. All of these revisions are marked in **[RED]** in the paper.
>
> ### **W1: Probing vs. Fine-Tuning in Audio SSL**
>  >*Statements such as the above make it seem like the audio SSL domain is not doing probing-based evaluation on frozen encoders at all. However, the paper only cites a handful of papers that support this notion, while ignoring the plethora of audio SSL works that evaluate frozen encoders using probing […]*
>
> **Our reasoning and statement.** First, thank you for providing this extensive list of papers. We agree that our initial phrasing was too broad and that the paper has to be more precise in that regard. It was not our intention to convey that linear probing has not been done at all in audio SSL. We have carefully revised the paper to clarify that our work addresses the specific line of research that chases SOTA performance on AudioSet (AS), where fully fine-tuning remains the dominant evaluation paradigm. In the following, we detail our reasoning:
>
>  - *SOTA on AudioSet:* Our work is primarily motivated by the line of work claiming SOTA on AS ( these include the benchmark’s models A-MAE, BEATs, ASiT, EAT SSLAM). All of these models, and similar works, almost exclusively compete based on fine-tuning results on AS.  This introduces confounding variables and reproducibility issues, making it difficult to assess the true quality of pretrained embeddings (see Introduction L. 47-48 and 50-55)
>  - *Lines of work:* Besides the SOTA AS line of work, your references show that a separate line of work utilizes (standard) probing, often on benchmarks like HEAR with a given evaluation suite. However, these two lines have remained largely separate as your mentioned works are only very rarely included in the result tables in the AS SOTA works. As a result, probing results from these models have had, unfortunately, limited influence on the SOTA race on AS. We believe that these models from, e.g., the HEAR track, might have achieved SOTA on AS if not for the "hyperparameter lottery" of fine-tuning, which obscures true model quality (the hyperparameters in EAT [2] and SSLAM [3] include a variety of factors such as augmentations, layer-wise lr decay, weighted selection etc. that can heavily impact rankings). This would potentially not have been the case if there was a principled way to compare the quality of pretrained models for AS.
>  - *Our contribution:* We discuss the problem of evaluating AS SOTA, provide a robust solution and an extensive analysis. Even among the good papers you cited, a systematic comparison of different probing techniques has not been done. Our method strongly differs from only performing linear probing on the cls-token or token averaging. We believe protobin closes the gap between finetuned and probed models enough to be considered as the first reliable solution for audio SSL model comparison, while avoiding the difficulties and reproducibility issues of finetuning on AS.
>  - *Validation:* One of your listed papers also highlights our motivation (Li et al. [3], 2024): *“Linear evaluation cannot fully reflect the capabilities of pre-trained models, as normally the models can be further fine-tuned with the data of downstream tasks. Fine-tuning experiments are conducted on the tasks of multi-label audio event classification (AS-2M, AS-20K and FSD50K) [...]”*. This is the exact problem we motivate and provide a solution for.
>
>  **Our actions based on your feedback.**
>    - We have very carefully revised the paper (Abtract L.12, Introduction L.50-55, 64-65, 77-78, 100, Related Work L.229-230, L.242-251, L.254-255, Conclusion L.532) to be precise about the "AudioSet SOTA" line of work we are addressing, removing any over-generalized statements.
>   - We have added a new part to Related Work in L.242-249, incorporating many of the papers you suggested. This new section explicitly discusses the two diverging evaluation paradigms and correctly positions our work.
>
> [1] Chen et al., "EAT: Self-Supervised Pre-Training with Efficient Audio Transformer", 2024.
>
> [2] Alex et al., "SSLAM: Enhancing Self-Supervised Models with Audio Mixtures for Polyphonic Soundscapes", 2024.
>
> [3] Li et al., "Self-Supervised Audio Teacher-Student Transformer for Both Clip-Level and Frame-Level Tasks", 2024.

---

> ### Author Response · Authors · 2025-11-17
> **Authors' Response (2)**
>
> ### **W2: Motivation**
> >*[…] However, several missing references listed in W1 are already aware of the drawbacks of conducting probing experiments only on [cls]-token or single vector representations obtained by pooling of information across all the patches.*
>
> **Our reasoning and statement.** We agree that awareness of the problem already exists, but we believe this strengthens the motivation for our work, rather than being a weakness. While prior works you mention identify the issue, our paper provides a novel solution (prototypes) and the first extensive, systematic benchmark with different probing methods in audio to analyze it. For instance, we show the differences in model rankings based on the evaluation method.
>
> >*[…] aggregation strategies that merge information across patches will inherently decimate frequency information. Niizumi et al. [2] were aware of this and proposed a simple strategy to aggregate temporal features while preserving frequency information.*
>
> **Our reasoning and statement**. We agree, as we hypothesize in the paper in the Introduction (L.67-68,82-92), that the size of the tokens and the aggregation method can create an information bottleneck in the context of utilizing frozen embeddings. However, we respectfully disagree that pooling information across patches directly decimates _all_ frequency information. Otherwise, weighted aggregation, such as cls-tokens, could also not succeed in fine-tuning. This could be caused by self-supervised objectives that diminish the selective property of tokens in self-attention modules (see Introduction Figure 1, L.67-68, and the performance improvement of linear with the additional supervised+ training, also discussed in Q4).
>
> Niizumi et al. [1] propose a smart idea to extract more information from the frozen embeddings by a simple reshaping method. We thank you for referring to this, it further supports our hypothesis: In the context of frozen embeddings, it rightly claims that averaging patch tokens discard useful information, and flattens the frequency patches into the feature dimension for every time token and averages the time axis. We agree this has advantages and can improve probing results over simple linear probing. However, this approach, as far as we know, has unfortunately[!] not been adopted by other work claiming SOTA on AS. Most importantly, our method is fundamentally different, providing another approach to the problem. While Niizumi et al. [1] cleverly reshape features, our method learns to disentangle features using multiple class-agnostic prototypes. Please also see the detailed answers to Reviewer [gW4z]’s comments on the method description.
>
> >*the <[cls]-linear probe> strategy is not representative of the entire audio SSL community's approach to evaluate audio SSL models, unlike what the paper in its current form makes it seem like.*
>
> **Our reasoning and statement.** We agree that a simple [cls]-probe does not represent the entire audio SSL field. However, looking through the papers you provided and prior work, we found that simple probing (on [cls] or averaged tokens), with the exception of the clever reshaping approach in Niizumi et al. [1], is indeed the most common in this line of work when fine-tuning is not used (also given HEAR’s fixed evaluation suite [2]). Our benchmark is the first to systematically compare these methods against more complex strategies like ours and attentive probing in audio SSL.
>
> **Our actions based on your feedback.**
>
> - **[NEW EXPERIMENT]** To directly compare our approach with prior work, we have implemented the reshaping method from Niizumi et al. (we refer to it as "linear preserve", linpre) and added it as a strong baseline in our benchmark. The updated results can be found in Table 2 in the paper. The description of the method is added in L.266-267, 323, 371.  In the following, we provide the new results of linpre in direct comparison with the results of our method, showing that it’s pretty competitive, despite its simple reshaping procedure.
> - While linpre offers a notable performance improvement over linear (+5.86 pp random averaged across backbones and datasets), protobin’s offers a more significant improvement of +13.42 pp over linear)
> - As mentioned above, we have updated the Related Work section to better position our method.
>
> [1] Niizumi et al., "Masked spectrogram modeling using masked autoencoders for learning general-purpose audio representation", 2022.
>
> [2] Turian et al., "Hear: Holistic evaluation of audio representations", 2022.

---

> ### Author Response · Authors · 2025-11-17
> **Authors' Response (3)**
>
> This comments shows the new results of linpre from Niizumi et al. We also add our protobin and the linear baseline as reference values:
>
> | Dataset | Backbone | linear | linpre | protobin |
> |---------|----------|---------|---------|----------|
> | **as20k** | **A-MAE** | 8.36 ± 0.0 | 16.49 ± 0.1 | 22.32 ± 0.1 |
> | | **ASiT** | 18.35 ± 0.0 | 18.53 ± 0.0 | 20.96 ± 0.0 |
> | | **Dasheng** | 20.98 ± 0.1 | 23.56 ± 0.0 | 29.94 ± 0.2 |
> | | **BEATs** | 24.71 ± 0.0 | 18.59 ± 0.0 | 31.54 ± 0.1 |
> | | **EAT** | 17.29 ± 0.0 | 26.49 ± 0.0 | 31.67 ± 0.1 |
> | | **SSLAM** | 17.04 ± 0.0 | 24.81 ± 0.0 | 30.94 ± 0.1 |
> | **fsd50k** | **A-MAE** | 19.71 ± 0.0 | 36.08 ± 0.1 | 49.69 ± 0.4 |
> | | **ASiT** | 39.57 ± 0.1 | 39.57 ± 0.1 | 46.70 ± 0.2 |
> | | **Dasheng** | 38.08 ± 0.2 | 45.11 ± 0.1 | 57.31 ± 0.0 |
> | | **BEATs** | 46.89 ± 0.0 | 39.93 ± 0.0 | 58.27 ± 0.2 |
> | | **EAT** | 36.39 ± 0.0 | 48.21 ± 0.1 | 55.64 ± 0.4 |
> | | **SSLAM** | 36.06 ± 0.0 | 46.11 ± 0.0 | 56.99 ± 0.1 |
> | **desed** | **A-MAE** | 57.46 ± 0.0 | 71.28 ± 4.2 | 85.57 ± 0.1 |
> | | **ASiT** | 72.92 ± 0.0 | 74.91 ± 0.1 | 81.74 ± 0.2 |
> | | **Dasheng** | 68.39 ± 0.0 | 74.49 ± 0.6 | 86.14 ± 0.3 |
> | | **BEATs** | 77.56 ± 0.0 | 76.97 ± 0.0 | 89.22 ± 0.6 |
> | | **EAT** | 76.15 ± 0.0 | 81.00 ± 0.0 | 88.82 ± 0.1 |
> | | **SSLAM** | 72.49 ± 0.0 | 80.31 ± 0.0 | 88.33 ± 0.3 |
> | **spass** | **A-MAE** | 58.94 ± 0.0 | 77.08 ± 0.2 | 79.95 ± 0.6 |
> | | **ASiT** | 68.80 ± 0.0 | 73.88 ± 0.0 | 74.69 ± 0.2 |
> | | **Dasheng** | 66.89 ± 0.0 | 72.07 ± 0.0 | 80.93 ± 0.5 |
> | | **BEATs** | 74.22 ± 0.0 | 82.12 ± 0.0 | 85.77 ± 0.4 |
> | | **EAT** | 65.96 ± 0.0 | 81.83 ± 0.0 | 85.64 ± 0.3 |
> | | **SSLAM** | 68.28 ± 0.0 | 80.74 ± 0.2 | 86.01 ± 0.1 |
> | **urban** | **A-MAE** | 58.72 ± 0.1 | 79.01 ± 0.1 | 85.17 ± 0.3 |
> | | **ASiT** | 77.53 ± 0.0 | 79.32 ± 0.0 | 82.28 ± 0.2 |
> | | **Dasheng** | 69.61 ± 0.1 | 77.30 ± 0.0 | 86.55 ± 0.1 |
> | | **BEATs** | 82.54 ± 0.1 | 81.61 ± 0.0 | 88.74 ± 0.2 |
> | | **EAT** | 77.76 ± 0.0 | 84.04 ± 0.0 | 89.24 ± 0.2 |
> | | **SSLAM** | 75.86 ± 0.0 | 79.01 ± 0.1 | 89.05 ± 0.4 |

---

> ### Author Response · Authors · 2025-11-17
> **Authors' Response (4/4)**
>
> ### **W3: Global average pooling**
>
> >*As mentioned in Weakness 2, Point 2. This pooling strategy across token positions is suboptimal and puts plain linear probe at a disadvantage for the models that do not expose cls tokens.*
>
> **Our reasoning and statement.** Thank you for pointing out this potential for confusion. To clarify, we do not use simple global average pooling in our benchmark at all. All models in our benchmark offer a cls-token that we utilize for simple linear probing. However, we also include more advanced pooling methods such as the attentive family that apply a type of weighted averaging across token positions and your proposed method, greatly improving the results. The cited text was only intended as a general explanation, and we see now that it was unclear.
>
> **Our actions based on your feedback.**
>
> - We have revised Section 2.2 to remove this ambiguity and ensure our actual benchmark methodology is clear in L142-144. Additionally, we added a small introductory sentence for this section in L.138.
>
> ### **W4: Miscellaneous statements**
>
> > […] These two papers are not representative of the entire audio SSL field. For e.g. [9, 13, 15] do not use a straightforward MIM style objective for learning audio representations through SSL.
>
> Thank you for this comment. We agree that our phrasing was imprecise. We meant that all SOTA AudioSet models investigated in our paper use a MIM-style objective [1-5] (also Dasheng even it if does not report the fine-tuning SOTA results on AS). We have revised the text to be more specific. While the works you have mentioned [6,7,8] also seem to use MIM-style objectives, we agree that it’s not as straightforward as the works we have mentioned. Particularly, [8] is based on distillation of already existing audio SSL models which might or might not use MIM-style objectives.
>
> **Our actions based on your feedback.**
>
> - We have rewritten the sentence in the Introduction about MIM-style objectives to be more precise in L.64-65 and in Related Work in L.232.
>
> > How does the paper on the BIRDNET dataset (Rauch et al., 2025b) justify that audio SSL defaults to using fine-tuning?
>
>  This was an error due to name similarity. The citation (BirdSet) was incorrect due to naming similarities and has been replaced by [1-5].
>
> **Our actions based on your feedback.**
>
> - We have corrected the wrong citation and replaced it by the the audio SSL model papers that utilize fine-tuning for AS (L.249).
>
> [1] Chen et al., "EAT: Self-Supervised Pre-Training with Efficient Audio Transformer", 2024.
>
> [2] Alex et al., "SSLAM: Enhancing Self-Supervised Models with Audio Mixtures for Polyphonic Soundscapes", 2024.
>
> [3] Huang et al., "Masked Autoencoders that Listen", 2022.
>
> [4] Dinkel et al., "Scaling up masked audio encoder learning for general audio classification", 2024.
>
> [5] Atito et al., “ASiT: Local-Global Audio Spectrogram Vision Transformer for Event Classification”, 2024.
>
> [6] Li et al., "Self-Supervised Audio Teacher-Student Transformer for Both Clip-Level and Frame-Level Tasks", 2024.
>
> [7] Schmid et al., "Effective Pre-Training of Audio Transformers for Sound Event Detection", 2025.
>
> [8] Chang et al., "USAD: Universal Speech and Audio Representation via Distillation", 2025
>
> ---
> ### **Conclusion**
>
> **We believe we have comprehensively addressed all your concerns, particularly regarding the paper's positioning and  the comparison to prior work. These revisions, based on your insightful feedback, have significantly strengthened the paper. If there are any remaining reservations that prevent you from recommending acceptance, please let us know, we are eager to address them.**

---

> ### Comment · Reviewer_FqWa · 2025-11-18
> **Good rebuttal. An issue regarding a plot.**
>
> Thank you for engaging in a fruitful rebuttal and for understanding my point of view, as well as making comprehensive changes to the language of the paper. The additional experiments on the linpre setting are very welcome. It has removed my reservations about the proposed approach.
>
> As I write this comment, I am going through all these changes in the paper as well as your rebuttals to other reviewers and once I have done that, I'll revisit my scores.
>
> In the meantime, I would like to point out an issue regarding Figure 6. In the figure, the protobin performance barplot for BEATs (70.71) appears to be identical to EAT (70.20) and smaller than SSLAM (70.26). I believe there is an error in this plot, maybe the printed BEATs average score is a misprint?

---

> > ### Author Response · Authors · 2025-11-18
> > **Authors' Quick Response**
> >
> > Dear **[FqWa]**,
> >
> > Thank you for your very quick and positive engagement in the discussion. We are very happy our revisions and the $\texttt{linpre}$ experiment have addressed your initial reservations.
> >
> > Regarding Figure 6: that is a very detailed and good catch. The numerical value for BEATs (70.71) is correct. The bar was simply rendered at an incorrect height. We have already corrected the figure in the paper and uploaded a new version. We appreciate your thorough review and willingness to revisit your initial score.

---

> > > ### Comment · Reviewer_FqWa · 2025-11-18
> > > **Thanks for clarifying, score updated.**
> > >
> > > Thanks for the clarification. After taking your rebuttal responses into account (responses to me as well as the other reviewers), I believe the updated paper is a significantly stronger submission. I have updated my score.

---

> ### Author Response · Authors · 2025-11-19
> **Thank you for the score update**
>
> Thank you very much for re-evaluating our work and for the score update to 8! We are very happy that the revisions and additional experiments have addressed your concerns. We truly appreciate the constructive criticism you provided, especially regarding the positioning of the paper and the inclusion of the $\texttt{linpre}$ baseline, as we are also convinced that this has made for a stronger and more robust submission.

---

### Official Review · Reviewer_gW4z · 2025-10-31

**Soundness:** 3
**Presentation:** 2
**Contribution:** 2
**Rating:** 4
**Confidence:** 4

**Summary:**

This paper primarily investigates the limitations of the existing linear evaluation paradigm for audio encoders.

The authors show that the use of [cls] tokens  for linear probing does not take into account the time-space localised representations contained in patch tokens, and this hurts downstream linear probing performance in audio encoders, especially given that most audio encoders employ reconstruction objectives where information is rich in the patch tokens.

The paper runs an evaluation benchmark on a set of linear probing methods on multiple downstream tasks, evaluated over multiple pre-trained audio encoders, and show that [cls] based methods perform poorly as opposed to learned pooling methods.

Motivated by this, the authors propose a pooling strategy (protobin) which adds to the existing method of prototype pooling by introducing binary constraints.

**Strengths:**

Clarity: The paper is well written and can be easily understood.

Motivation: The paper is well motivated, and the motivation is justified by the results in Appendix A, where [cls] tokens alone perform much worse than learned pooling methods.

Contribution 1: The paper presents a benchmark and evaluation of recent SSL methods under multiple pooling/aggregation strategies. Using this benchmark, SSL methods on audio data can be fairly evaluated without a bias towards the [cls] token, thereby nullifying the apparent advantage seen by discriminative-only based methods on linear probing performance.

Contribution 2: The results by using the binarized prototype tensor (protobin and proto combined) yield SoTA  linear probing performance across almost all tasks and pre-trained models. Therefore, prototype-based aggregation can act as a standard method for performing linear probing for multi-label audio classification.

**Weaknesses:**

Significance: The fact that [cls] tokens alone perform worse than learned pooling mechanisms is already mentioned in literature [1]. Therefore, the same message being repeated for audio needs a stronger justification, with possible reasons as to why the message in [1] might or might not hold for audio data, thereby justifying the motivation for evaluation.

Originality: The protobin aggregator is simply a binarization of the prototype tensor derived from Bird-MAE. Looking at Table 4 and 5, proto and protbin share similar overall performance, with protobin performing marginally better in more tasks than protobin (approximately 2/3), but not significantly enough to justify the addition of binarization as a contribution. The actual performance gain over other methods seems to be gained by the use of prototypes, as in Bird-MAE [2].

The paper does not clearly explain why the cosine similarities between prototypes are mostly zero after training, despite the absence of any explicit mechanism to control the prototype distribution. It is unclear how the \(\pm{1}\) constraint alone enforces this near-orthogonality or prevents different prototypes from converging to similar representations.

Multi-label classification as a standalone downstream task: Although the extension towards other tasks is mentioned in future works, having multi-label classification alone as the evaluation framework does not provide a full picture on leveraging information from frozen features.

Minor: Some of the formatting seems to be too cluttered, for example the contributions and Q sections in results. It takes away from the clarity of the paper rather than adding to it. However, I acknowledge that this is subjective.



[1] Przewięźlikowski, Marcin, et al. "Beyond [cls]: Exploring the true potential of Masked Image Modeling representations." Proceedings of the IEEE/CVF International Conference on Computer Vision. 2025.

[2] Rauch, Lukas, et al. "Can Masked Autoencoders Also Listen to Birds?." arXiv preprint arXiv:2504.12880 (2025).

**Questions:**

Control of prototype distributions:
How are the cosine similarities of prototypes mostly zero after training (Figure 4) if there is no explicit method to control the distribution?
What is preventing two prototypes from learning the same thing? More specifically, why does the \(\pm{1}\) constraint force near-0 similarity? It is not clear in the paper, and how this prototype-to-class contributions emerge is not explained in the results.

Why exactly does the binarization work better in some tasks and not in others?

---

> ### Author Response · Authors · 2025-11-17
> **Authors' Response (1)**
>
> Dear **[gW4z]**,
>
> First of all, thank you for your thoughtful and constructive feedback. We appreciate that you recognized our paper’s key strengths, including clarity, motivation and novel benchmark, and our finding that prototype-based pooling achieves SOTA performance. We have found your suggestions very helpful for improving the paper. Below, we address each of your concerns, summarizing our response and the concrete actions taken. We have very carefully and thoroughly revised parts of the paper to implement your feedback, all changes based on your feedback are marked in **[BLUE]** in the revised paper.

---

> ### Author Response · Authors · 2025-11-17
> **Authors' Response (2)**
>
> ### **W1: Significance**
> > *The fact that [cls] tokens alone perform worse than learned pooling mechanisms is already mentioned in literature [1]. Therefore, the same message being repeated for audio needs a stronger justification, with possible reasons as to why the message in [1] might or might not hold for audio data, thereby justifying the motivation for evaluation.*
>
> **Our reasoning and statement.** Thank you for this crucial point. We agree that our motivation would benefit from a stronger separation from prior work in computer vision by Przewięźlikowski et al. [1] We have revised the paper to clarify that our work is not simply confirming their findings in a new domain, but rather building upon computer vision insights to address a more nuanced problem in audio that requires a different solution. Our work shifts the conversation from only "the [cls] token is flawed" (which we agree is established) to "what is a faithful way to aggregate tokens for the unique challenges of audio SSL," and we provide extensive evidence that prototypical probes are an adequate answer.
>
> Our core contribution is demonstrating that while a form of attentive pooling (the solution in [1]) is an improvement, a fundamentally different approach, (multi-vector prototypical pooling) is required to fully unlock the potential of audio embeddings. Specifically, our work provides the following novel insights that distinguish it from prior computer vision research:
>
> 1. *Fine-tuning*: While computer vision research has questioned the [cls]-token (as you mentioned [1] and other papers we mention in L.59-60), costly fine-tuning remains the standard for achieving SOTA results in AudioSet (Audio-MAE→BEATs→ASiT→ EAT→SSLAM). Although the difficulty of achieving reasonable performance is mentioned in audio SSL [2], a reliable solution and analysis is missing. The motivation for our paper is to provide a reliable method for a fair comparison of audio SSL models, while removing confounding factors stemming from fine-tuning (e.g., augmentations, hyperparameters etc.), which has made reproducibility an issue in the mentioned line of work (L.50-55, Introduction).
> 2. *Domain differences*: Although many methods are very similar and are transferable across data modalities, there are technicalities that make application from other modalities challenging. For example, equivalent SSL models between domains (e.g., Audio-MAE [2] and Image-MAE [3]) differ in quality of frozen embeddings - it is more challenging to get good performance in audio: Audio-MAE did not report any frozen embedding performance in the paper for a reason (see our results). A reason could be higher density of information in natural images compared to that of an audio spectrogram. The method in [1] is a simple weighted averaging of tokens (comparable to attentive probing mechanisms in our benchmark). However, the audio SSL embeddings seem to require more work due to higher entanglement of tokens (see Figure 1). Our approach is not only an aggregation of tokens, it’s an aggregation combined with mechanisms that support automatic disentanglement of information per classes (L.105-107, Introduction).
> 3. *Our analysis*: We go far beyond attentive pooling. Our comprehensive benchmark is the first in audio to systematically compare attentive methods against prototypical methods for SOTA  AudioSet models. For instance, as you can see in Figure 6, our method has revealed the superiority of older SSL audio models (e.g., BEATs) compared to recent ones like Dasheng (despite being trained on almost 50 times more data [4]). This shows the inadequacy of fine-tuning for assessing the quality of SSL audio model.
>
> **Our actions based on your feedback.**
>
> - We have rewritten Contribution 1 to focus on our unique questions (Q1-Q4) in audio SSL in L.94-98, which goes beyond prior work in computer vision. These include the method hierarchy (Q1), if linear is a faithful evaluator (Q2), multi- vs. single-label differences (Q3), and the impact of an extra supervised adaption step before probing (Q4). This extensive analysis is not done in [1].
> - We have revised the Introduction (L. 70-71), the "Positioning of this work" section (L. 286-291) and added an extra sentence to “Advanced probe architectures” (L.275) to explicitly position our work.
> - We added the year of publication to Figure 6 to emphasize how our superior probing method re-ranks models, showing the inadequacy of both simple linear probes and fine-tuning for assessing the true quality of SOTA audio models, a practical problem not present/adressed in the vision literature.
>
> [1] Przewięźlikowski et al., "Beyond [cls]: Exploring the true potential of Masked Image Modeling representations”, 2025.
>
> [2] Huang et al., "Masked Autoencoders that Listen", 2022.
>
> [3] He et al., "Masked Autoencoders Are Scalable Vision Learners", 2022.
>
> [4] Dinkel et al., "Scaling up masked audio encoder learning for general audio classification", 2024.

---

> ### Author Response · Authors · 2025-11-17
> **Authors' Response (3)**
>
> ### **W2: Originality (contribution)**
> > *Originality: The protobin aggregator is simply a binarization of the prototype tensor derived from Bird-MAE. […] but not significantly enough to justify the addition of binarization as a contribution. The actual performance gain over other methods seems to be gained by the use of prototypes, as in Bird-MAE [2].*
>
> **Our reasoning and statement.** Thank you for this insightful comment. We have revised the paper to clarify that protobin's primary contribution is not a minor performance gain, but a significant architectural simplification, generalization, and efficiency improvement over prior work. Our goal was to design a simpler, more efficient, and elegant probing method for fair SSL evaluation (as one part of the contributions of the paper). While we agree that the main performance leap comes from the prototypes, protobin introduces two main architectural simplifcations that further justify its contribution:
>
> 1. *Decoupled prototypes:*  Unlike prior work [1], our prototypes are class-agnostic. Their specific class contributions are learned automatically by the final linear layer, making the method more flexible and removing design constraints. This has the potential to reduce the number of required prototypes because it allows the prototypes to collaborate in disentangling the necessary information for the task (see additional Ablation study later).
> 2. *Removing orthogonality loss:* We found that the supervised training, due to quality of human annotation, is enough to motivate a diverse set of prototypes. Hence, removing this orthogonality loss term further simplifies the method compared to prior work [1] without a performance trade-off.
>
> These simplifications do not come at a performance cost. Our results show that the simpler protobin consistently matches or exceeds the performance of the more complex proto baseline. We argue that achieving superior results with a simpler method is a very valuable contribution to evaluating SOTA SSL models in audio to more reliably assess frozen embeddings  (see Contribution 2, L.101-103).  Furthermore, our contributions for this method are much broader:
>
> 1. *Efficiency*: Binarization offers a potential 32x memory reduction for the prototypes in application settings, which is a significant step toward making probing a truly lightweight alternative to fine-tuning (e.g., for on-device in bioacoustic tasks).
> 2. *Beyond the method*: As far as we know, our work is the first to elevate prototypical probing from a niche bioacoustics methods into a general evaluation paradigm for frozen representations in audio. Through the first comprehensive benchmark in audio SSL probing, we demonstrate its effectiveness across diverse, general-purpose audio SSL backbones and tasks, providing a robust evaluation tool.
>
> **Our actions based on your feedback.**
>
> - We have rephrased protobin’s contribution in the contributions box to emphasize simplification and efficiency (Introduction L.104-107).
> - We have reordered contributions to better highlight the importance of elevating probing as a standard evaluation method.
> - We have revised the "Positioning of this work” paragraph (L. 281-285) to more clearly articulate protobin's novelty.
>
> [1] Rauch et al., "Can Masked Autoencoders Also Listen to Birds?", 2025.

---

> ### Author Response · Authors · 2025-11-17
> **Authors' Response (4)**
>
> ### **W2: Originality (method explanation)**
> > *The paper does not clearly explain why the cosine similarities between prototypes are mostly zero after training, despite the absence of any explicit mechanism to control the prototype distribution. It is unclear how the (\pm{1}) constraint alone enforces this near-orthogonality or prevents different prototypes from converging to similar representations.*
>
> **Our reasoning and statement**. Thank you for pointing this out, this enables us to improve the method’s description. We empirically observe that the near-orthogonality emerges naturally from two factors: the implicit pressure from the supervised loss and the structural bias of the binarization constraint, which we now clarify in the paper.
>
> The supervisory signal is sufficient to promote diversity of prototypes, preventing the protoypes from converging to similar representations. The binarization further aids this by creating a geometric bias towards solutions with large angular margins, making it unlikely for prototypes to converge on redundant representations. However, the binarization is primarily done for the purpose of efficiency for future applications in audio, for example, employing on-device applications.
>
> **Our actions based on your feedback.**
>
> - We have added a clearer explanation of the prototype diversity mechanism to the Methodology section in L.179-181 and revised the Rationale in L.203-215.
>
>
> ### **W2: Originality (task)**
> > *Multi-label classification as a standalone downstream task: Although the extension towards other tasks is mentioned in future works, having multi-label classification alone as the evaluation framework does not provide a full picture on leveraging information from frozen features.*
>
> **Our reasoning and statement.** We agree that evaluating on multi-label tasks alone would be insufficient. To provide a comprehensive picture, our benchmark already includes single-label (ESC-50, SC) and few-shot, domain-shifted (BirdSet) tasks as control tasks (Appendix A, Table 5 and Table 6). This design allows us to assess the capabilities of the AudioSet-pretrained models across a range of conditions. As our focus is on general audio classification tasks that are used in the line of audio SSL work we refer to, we believe this additional task ablation is sufficient for supporting our claims. We are not certain what other tasks you have in mind. We agree that exploring the utility of SSL audio embeddings in other areas is a valuable future direction.
>
> **Our actions based on your feedback.**
>
> - To make the purpose of these control tasks more explicit, we added a clarifying sentence to our dataset description method in L. 314-318
>
> ### **Formatting**
> >*Some of the formatting seems to be too cluttered, for example the contributions and Q sections in results. It takes away from the clarity of the paper rather than adding to it. However, I acknowledge that this is subjective.*
>
> We appreciate your feedback. We had to make a decision regarding the trade-off between density and completeness given the page constraints. We decided that all the included material was necessary to fully support our contributions. Therefore, while we have incorporated all other suggestions into the revised paper, we have kept the current presentation.

---

> ### Author Response · Authors · 2025-11-17
> **Authors Response (5)**
>
> ### **Questions**
> >*Control of prototype distributions: How are the cosine similarities of prototypes mostly zero after training (Figure 4) if there is no explicit method to control the distribution? What is preventing two prototypes from learning the same thing? More specifically, why does the (\pm{1}) constraint force near-0 similarity?*
>
> >*It is not clear in the paper, and how this prototype-to-class contributions emerge is not explained in the results.*
>
> Thank you for these excellent follow-up questions. As they are very similar to the weaknesses we have already addressed above, we believe we answered all of them in the previous comments in detail. If there is still something unclear, please let us know.
>
> > *Why exactly does the binarization work better in some tasks and not in others?*
>
> As a clarification for your last question: the performance differences are not a consequence of only the binarization, per se. It’s also due to the decoupling of the prototypes, which allows all of them to collaborate while the prototypes in prior work were restricted to specific classes. The additional benefits of binarization are twofold: (i) simplifying the probing method so that the results are more faithful to the genuine quality of frozen embeddings, (ii) reduce computation cost for potential on-device applications. Regarding the performance differences between single-label and multi-label tasks: the information bottleneck from the pretrained models is less severe in multi-class. A strong single-vector method like mhca can be sufficient by focusing on the one dominant sound, which is why the unique advantage of our multi-vector approach is less pronounced there. In contrast, multi-label polyphonic audio makes this bottleneck critical. Single-vector methods fail, whereas our prototypical approach excels by using different prototypes to represent different sound events simultaneously.
>
> **Our actions based on your feedback.**
>
> - As stated above, we added a detailed explanation to the Rationale Section of the paper (L.203-215).
> - We improved the Q3 Takeaway to better explain why prototypical methods excel in multi-label scenarios (L.468-474).
> - [NEW EXPERIMENT] We added a new detailed ablation study (Appendix B: Ablation Study, L.894-983, Table 7) comparing different numbers of prototypes and a decoupled, non-binarized variant (protofloat) to better isolate the performance effects of our design choices.
> - We see that a full 32-bit precision can capture finer-grained acoustic details, which gives it a slight performance edge in some configurations (e.g., in the urban dataset with more polyphonic sounds).
> - We have added a discussion of this trade-off from proto to protobin to the Q1 results section (L.376-381).
>
> -----
>
> ### **Conclusion**
> **We believe these revisions, guided by your thoughtful feedback, have significantly improved the paper's methodological clarity and impact. Thank you again for your time and expertise. We are convinced we have thoroughly addressed all your concerns. If any reservations for accepting the paper remain, we would be happy to address them.**

---

> ### Author Response · Authors · 2025-11-17
> **Authors' Response (6/6)**
>
> In this comment, we provide the new ablation study (Appendix B, Table 7):
>
> | Backbone | Probe | as20k (J=1) | as20k (J=5) | as20k (**J=10**) | as20k (J=20) | urban (J=1) | urban (J=5) | urban (J=10) | urban (**J=20**) | esc50 (J=1) | esc50 (J=5) | esc50 (J=10) | esc50 (**J=20**) |
> |:---|:---|---:|---:|---:|---:|---:|---:|---:|---:|---:|---:|---:|---:|
> | A-MAE | protobin | 20.14 | 21.91 | 22.32 | 22.40 | 73.64 | 83.80 | 84.67 | 85.17 | 55.92 | 78.92 | 81.33 | 83.70 |
> | A-MAE | protofloat | 20.87 | 22.55 | 23.01 | 23.07 | 77.46 | 84.69 | 85.37 | 86.01 | 67.67 | 80.00 | 81.75 | 82.10 |
> | A-MAE | proto | 19.08 | 21.05 | 21.61 | 21.95 | 64.03 | 83.12 | 83.02 | 83.63 | 49.75 | 73.92 | 77.25 | 82.59 |
> | A-MAE | linear | 8.36 | 8.36 | 8.36 | 8.36 | 58.72 | 58.72 | 58.72 | 58.72 | 22.08 | 22.08 | 22.08 | 22.08 |
> | BEATs | protobin | 26.70 | 27.68 | 31.54 | 31.93 | 78.23 | 87.25 | 88.12 | 88.74 | 69.50 | 82.08 | 83.25 | 84.10 |
> | BEATs | protofloat | 27.64 | 30.52 | 30.89 | 31.68 | 79.90 | 87.92 | 88.60 | 88.63 | 75.75 | 84.25 | 84.58 | 84.70 |
> | BEATs | proto | 27.63 | 30.47 | 30.54 | 30.66 | 77.41 | 87.55 | 88.64 | 89.04 | 77.25 | 83.67 | 84.67 | 85.08 |
> | BEATs | linear | 24.71 | 24.71 | 24.71 | 24.71 | 82.54 | 82.54 | 82.54 | 82.54 | 78.92 | 78.92 | 78.92 | 78.92 |
> | ASiT | protobin | 20.74 | 20.21 | 20.96 | 21.71 | 78.59 | 81.35 | 81.87 | 82.28 | 75.08 | 79.25 | 80.17 | 80.30 |
> | ASiT | protofloat | 21.18 | 21.57 | 21.21 | 21.30 | 79.21 | 82.09 | 82.48 | 82.19 | 76.00 | 79.25 | 79.75 | 80.12 |
> | ASiT | proto | 21.31 | 21.94 | 21.89 | 20.73 | 68.89 | 81.20 | 81.89 | 82.35 | 73.50 | 78.75 | 81.95 | 82.44 |
> | ASiT | linear | 18.35 | 18.35 | 18.35 | 18.35 | 77.53 | 77.53 | 77.53 | 77.53 | 76.08 | 76.08 | 76.08 | 76.08 |
> | EAT | protobin | 29.08 | 31.61 | 31.67 | 32.12 | 80.05 | 88.60 | 89.01 | 89.24 | 71.00 | 84.25 | 85.00 | 86.81 |
> | EAT | protofloat | 29.11 | 31.04 | 31.19 | 31.81 | 82.00 | 88.90 | 89.08 | 89.14 | 82.58 | 87.58 | 88.25 | 89.14 |
> | EAT | proto | 28.64 | 30.65 | 31.06 | 31.30 | 72.34 | 87.65 | 88.84 | 89.11 | 55.33 | 78.08 | 82.58 | 85.91 |
> | EAT | linear | 17.29 | 17.29 | 17.29 | 17.29 | 77.76 | 77.76 | 77.76 | 77.76 | 75.33 | 75.33 | 75.33 | 75.33 |
> | SSLAM | protobin | 28.69 | 29.77 | 30.94 | 32.10 | 81.08 | 86.82 | 88.92 | 89.05 | 65.75 | 80.67 | 83.83 | 84.70 |
> | SSLAM | protofloat | 29.08 | 30.50 | 30.55 | 31.26 | 81.63 | 87.16 | 89.05 | 89.05 | 79.75 | 86.00 | 86.17 | 86.69 |
> | SSLAM | proto | 29.08 | 30.53 | 30.84 | 30.99 | 81.60 | 88.58 | 88.45 | 88.82 | 62.08 | 80.67 | 82.17 | 85.18 |
> | SSLAM | linear | 17.04 | 17.04 | 17.04 | 17.04 | 75.86 | 75.86 | 75.86 | 75.86 | 74.17 | 74.17 | 74.17 | 74.17 |

---

### Author Response · Authors · 2025-12-02
**Authors' General Comment and Summary of Re-Evaluation (1/2)**

Dear AC, Dear reviewers,

First, we would like to thank the reviewers again for the feedback and the very productive discussion during the shortened rebuttal. Second, we also want to thank you, the AC, for taking your time for the rest of the process. We are particularly pleased that our revisions and additional experiments addressed all reviewers’ concerns.  **This led to a very positive consensus on the paper’s value for the community with a new intermediate mean score of 6.67 (8,8,4)**.

While the revisions based on the rebuttal have improved the paper, the core contributions and strengths for the audio SSL community, acknowledged by the reviewers, remain intact:

- **Novel and extensive benchmark**: We conduct the first extensive benchmark in AudioSet-based SSL for frozen representation. This was recognized “*comprehensive and rigorous*” [HEBB], “*thorough, with a good selection of audio SSL models*” [FqWa], and “*using this benchmark, SSL methods on audio data can be fairly evaluated without a bias towards the [cls] token*” [gW4z].
- **Elevating probing in audio SSL**:  We demonstrate how different probing techniques (e.g., attentive, linear, prototypical) change model rankings, highlighting the limitations of the current fine-tuning evaluation paradigm when chasing SOTA on AudioSet. Reviewers noted the problem analysis as *"systematic and insightful"* [HEBB], the motivation as *"justified by the results"* [gW4z], and conclude "*prototype-based aggregation can act as a standard method for [...] probing*" [gW4z].
- **New prototypical method ($\texttt{protobin}$)**: We introduce a novel binarized prototypical probing method that simplifies and improves upon prior methods. We propose class-agnostic prototypes with a class-wise information-aggregation to efficiently utilize frozen representations. It was praised to have *"methodological elegance and efficiency"* [HEBB], shown to *"work well"* [FqWa] and “[…] *using the binarized prototype tensor yield SoTA linear probing performance across almost all tasks and pre-trained models”* [gW4z].

In the following comment, we summarize the key outcomes of the extensive rebuttal, the improvements made to the paper, and the resulting
$\color{red}{\text{re-evaluation, leading to a very positive intermediate score distribution of (8,8,4).}}$📌

---

> ### Author Response · Authors · 2025-12-02
> **Authors' General Comment and Summary of Re-Evaluation (2/2)**
>
> ### **Major Re-Evaluation: From Reject (2) to Clear Accept (8) [Reviewer FqWA]**
>
> We successfully addressed the concerns regarding the positioning of the paper and the comparison to prior work raised by [FqWA].
>
> **Action**: We clarified the distinction between our work and the “AudioSet SOTA” fine-tuning paradigm with additional literature. We also added a new baseline experiment ($\texttt{linpre}$ from Niizumi et al. [1]) to the benchmark (Table 2 and markdown table in the detailed rebuttal) **.** Changes are marked in [RED] in the paper, changes for multiple reviewers in [GRAY].
>
> **Outcome:** These changes and clarifications led to a significant change in the reviewer’s assessment, raising their score from a $\color{red}{\text{2 to an 8 with the maximum level of confidence}}$. This improved the paper’s scoring significantly:
> > Reviewer FqWa (18 Nov 2025): *“**Thank you for engaging in a fruitful rebuttal** and for understanding my point of view, as well as making comprehensive changes to the language of the paper. The additional experiments on the linpre setting are very welcome. **It has removed my reservations about the proposed approach."***
>
> > Reviewer FqWa (18 Nov 2025): *“After taking your rebuttal responses into account (responses to me as well as the other reviewers), **I believe the updated paper is a significantly stronger submission. I have updated my score.”***
>
> [1] Niizumi et al., "Masked spectrogram modeling using masked autoencoders for learning general-purpose audio representation", 2022.
> ___
>
> ### **Comprehensive Addressal of Concerns (4) [Reviewer gW4z]**
>
> Unfortunately, [gW4z] has not yet respond to our rebuttal in the shortened period. However, we are confident that we have thoroughly addressed all raised weaknesses. While the reviewer acknowledged our contributions (prototypical probes as a strong evaluation tool) and clear motivation, some concerns were raised regarding the justification of our method compared to previous work and the clarity of its mechanisms.
>
> **Action:** We revised small parts of the text to explicitly differentiate our contribution from prior computer vision work, emphasizing the unique reliance on fine-tuning and absence of probing when chasing SOTA on AudioSet with SSL. Our new Ablation Study (Appendix B, Table 7 and in the detailed rebuttal as a markdown table), which were positively noted by [HEBB], directly addresses this reviewer’s query about the justification of our method. Additionally, we added a rationale in the methodology section explaining how the supervised loss and binarization constraints naturally enforce diversity without explicit orthogonality terms from previous work. Changes are marked in [BLUE] in the paper.
>
> **Outcome:** While there is no response yet, the very extensive reasoning and the fact that these clarifications are related to the fully resolved concerns of [HEBB] as well as the inclusion of our response in [FqWa]'s re-evaluation give us confidence that the points are fully addressed.
>
> ___
>
>
> ### **Confirmed Clear Accept (8) [Reviewer HEBB]**
>
> The reviewer was very positive and acknowledged the strengths of our work. We successfully addressed the requests for deeper methodological analysis regarding prototype sensitivity and binarization trade-offs mentioned.
>
> **Action**: We introduced a comprehensive ablation study (Appendix B, Table 7 and in the detailed rebuttal as a markdown table), analyzing the impact of number of prototypes and compares protobin against a non-binarized version protofloat to isolate the effects of our binarization. Changes in the paper are marked in [GREEN] and [GRAY] in the appendix.
>
> **Outcome**: The reviewer confirmed that their concerns were fully resolved and maintained their clear acceptance rating of 8.
>
> > Reviewer HEBB (24 Nov 2025): “*I would like to thank the authors for their extensive rebuttal. The **clarifications provided were very helpful in resolving my initial queries.** I will be **maintaining my recommendation** and **hope to see this work presented at the conference**.*”
>
> ___
>
> ### **Conclusion**
> The rebuttal process has resulted in a stronger paper while preserving the core message and contributions that were praised by the reviewers. With an additional probing baseline, detailed ablations, a clearer method description and more precise positioning, we have addressed all reviewers’ concerns. **With [FqWa] moving to a clear accept (8), [gW4z]'s concerns addressed in detail (4), and [HEBB] confirming their strong support (8), we believe the paper delivers a very valuable contribution to the audio SSL community.**

---

### Meta-Review · Area_Chair_DnLT · 2025-12-30

**Summary:**

The original submission made too broad claims and did not clearly explain the motivation and the contribution of the work. All available reviews are insightful, as are the rebuttals. The criticism is addressed in the revised version, which brought one reviewer from reject to clear accept (where an existing reviewer already is). A third reviewer did not update their score, but the rebuttal is substantive and it seems consistent that the reviewer would raise their score.

**Reviewer Concerns:**

I think all reviewer concerns should have been addressed in the revised submission, since the rebuttal was substantive and acknowledged by (some) reviewers to resolve their questions and concerns in discussion.

gW4z did not respond, but authors submitted substantive rebuttals, and the rebuttal seems to address the reviewers questions, especially wrt significance and originality, which are largely clarification questions IMO. Some "minor" formatting questions may remain, but the updated paper looks good to me.

**Reviewer Scores:**

FqWa: 2 -> 8 (confirmed according to the review, this seems to be a genuine improvement, before any leaking)
gW4z: 4 -> ~6 (seems reasonable, since gW4z's concerns should also be addressed)
HEBB: 8 (stays)

---

### Decision · Program_Chairs · 2026-01-26

Accept (Poster)